# The contribution of tourism mobility to tourism economic growth in China

**Jun Liu[1], Mengting Yue[2], Fan Yu[3]\*, Yun Tong[4]**

**1** School of Tourism, Hubei University, Wuhan, Hubei, China, **2** School of Urban and Regional Science, East China Normal University, Shanghai, China, **3** School of Business, Hubei University, Wuhan, Hubei, China, **4** School of Tourism, Hainan University, Haikou, Hainan, China

\* 201901111200032@stu.hubu.edu.cn

**Data Availability Statement:** The data on RAILWAY, HIGHWAY, ROAD1, ROAD2, ROAD, GDP, TERTIARY INDUSTRY, and POPULATION are from the Chinese Nation Bureau of Statistics (https://data.stats.gov.cn/easyquery.htm?cn=C01).

## Abstract

Mobility is the key factor in promoting tourism economic growth (TEG), and the transportation infrastructure has essential functions for maintaining an orderly flow of tourists. Based on the theory of fluid mechanics, we put forward the indicator of tourism mobility (TM). This study is the first to measure the level of TM in China and analyze the spatiotemporal evolution characteristics of TM. Applying the Exploratory Spatial Data Analysis method, we analyze the global and local spatial correlation characteristics of TM. Moreover, we further estimate the contribution of TM to TEG by econometric models and the LMDI method. The results show that (1) the TM in China has maintained rapid growth for a long time. However, there are differences in the rate of growth in different regions. The TM in each region only showed a significant positive spatial correlation in 2016–2018. The space-time pattern is constantly changing over time. The local spatial autocorrelation results of TM are stable, and various agglomeration states are stably distributed in some provinces. (2) The regression results of the traditional panel data model and spatial panel data model both show that TM has a significant positive effect on TEG. Moreover, TM has a negative spatial spillover effect on neighboring regions. (3) The result from the decomposition of LMDI shows that the overall contribution of TM to TEG is 15.76%. This shows that improving TM is a crucial way to promote the economic growth of tourism.

## Introduction

In recent years, the tourism industry has maintained rapid development. By 2019, the total number of global tourist trips exceeded 12.3 billion, an increase of 4.6% over the previous year. The total global tourism revenue was US$5.8 trillion, equivalent to 6.7% of global GDP (World Tourism Economy Trends Report [1]). Tourism has made important contributions to economic growth by increasing employment, improving infrastructure, and accumulating foreign exchange earnings for destinations [2]. Due to the impact of COVID-19, People's travel is restricted. The total number of international tourists in 2021 decreased by 72% compared with 2019, and international tourism consumption dropped by nearly half compared with 2019 [3].

The above facts remind us that mobility has become an essential feature of tourism activities [4, 5]. Tourists from origins to destinations result in a series of mobility of information,

The data on TOURISM REVENUE and VISITORS are from the CEIC database (https://insights. ceicdata.com). The data on TRAFFIC, TOURISM MOBILITY, RECPTION, INDUSTRY, and STRUCTURE were calculated by the authors. Please see the paper for details.

**Funding:** This work was supported by grants from National Social Science Foundation of China [grant number 17CJY051].

**Competing interests:** The authors have declared that no competing interests exist.

material, and capital. These mobilities have a great influence on tourist destinations [6–9]. If tourism mobility (TM) stagnates, tourist attractions, reception facilities and transportation facilities built for tourists will be idle. Tourism workers will lose their jobs and tourism economic growth (TEG) will also stagnate. Therefore, studying the impact of TM is necessary and important.

As one of the important tourist destinations in the world, China's domestic tourism and inbound tourism are developing rapidly. In 2019, the total contribution of China's tourism industry to GDP reached 10.94 trillion yuan, accounting for 11.05% of the total GDP, exceeding the proportion of international tourism in the global GDP. A total of 28.25 million people were directly employed in tourism, and 51.62 million people were indirectly employed in tourism. The total employment in tourism accounts for 10.31% of the total employed population in the country [10]. However, due to the impact of COVID-19, the development level of China's tourism industry has not recovered to the level of 2019. In 2021, the total number of domestic tourists in China was 3.246 billion, which is only 54% of that in 2019, and directly leads to a total tourism revenue of 2.92 trillion yuan, which is only 51% of that in 2019. This shows that TM is more important to China's tourism industry. Therefore, we decide to focus on the TM in this study and take China as the research sample.

The top priority of this study is to obtain the right measurement of TM. Transportation infrastructure is an important carrier for the exchange of factors in tourism. Existing studies have confirmed that transportation is a key factor in promoting TEG [11–13]. The establishment of the transportation system has an obvious effect on improving the accessibility of tourist destinations and promoting the inflow of the tourist population [14]. However, most existing studies only take tourist arrivals to characterize TM [15–21]. They ignore that the transportation infrastructure is also an important factor affecting the TEG. Therefore, this study redefines TM, which considers both transport infrastructure and tourist arrivals.

Another important purpose of this study is to explore the effect of TM on TEG. Existing literature analyzes the links between TM and international trade [22, 23] or focuses on the relationship between economic growth [24, 25]. However, less literature has focused on the relationship between TM and TEG. There are two possible reasons for the lack of attention. First, the positive and significant impact of the tourist arrivals and TEG no longer needs to be verified. It is common sense that the more tourists the destination receive, the higher the tourism income. Second, tourist arrivals, as a single indicator to measure TM, are able to affect the TEG. Our measurement of the TM concludes both transport infrastructure and tourist arrivals in this study. Therefore, we decide to explore the contribution of TM to the TEG based on the new measurement for TM.

We first use econometric methods to test whether there is a significant impact of TM on TEG. Considering the positive impact of transport infrastructure on China's TEG [26], we hypothesize that TM has a positive impact on TEG. Previous studies have also shown that the spatial spillover effect of tourism may significantly affect the TEG [27–29]. Therefore, we further apply the spatial Durbin model to test the impact of TM on TEG.

Moreover, we also use the LMDI (Logarithmic Mean Divisia Index) method to further analyze the contribution of TM to TEG in more detail. The LMDI method is often used to study environmental issues such as energy consumption and carbon emissions [30, 31]. In the field of tourism research, the LMDI method is mostly used to decompose tourism carbon emissions or energy consumption [32, 33]. Few studies are using the LMDI to analyze TEG. Therefore, we further use the LMDI method to decompose TEG into five influencing factors including the tourism mobility effects (*TM*), the cumulative traffic effects (*Traffic*), the effects of the tertiary industry (*Industry*), the structural effects of the tourism industry (*Structure*) and the reception effects (*Reception*), and examine the contribution of TM to TEG.

Different from previous studies, this study makes two contributions to the literature. First, we introduce the related concepts of fluid mechanics to construct the indicator TM. We also consider the superposition effect of tourist arrivals and transportation infrastructure. This deepens the understanding of TM and promotes the integration of interdisciplinary knowledge. Second, we are the first to examine the impact of TM on TEG using econometric models and the LMDI method. This deepens the understanding of the mechanisms that influence TEG. The results of this study also provide a reference for tourism-related policy makers. Regions wishing to develop tourism can achieve TEG by expanding the size of the source market and promoting the construction of transportation infrastructure.

The rest of this study is organized as follows. Section 1 summarizes the relevant literature. Section 2 presents the theoretical framework, methods, and data. Section 3 introduces the spatiotemporal pattern and evolutionary trend of TM. Section 4 analyzes the contribution of TM to TEG from two different perspectives. Section 5 discusses and analyzes the research results. The last section concludes this study.

## Literature review

As the core of tourism activities, TM refers to the mobility of tourists from the origin to the destination, and the stay of tourists in the region [34]. It is often associated with tourism demand and is measured by tourist arrivals [35]. Since the 1970s, many studies have paid attention to the influencing factors and the spatial structure of TM [15, 16]. The existence of regional heterogeneity makes TM affected by many factors, such as infrastructure, income, GDP, and cultural distance [17, 18, 20]. Moreover, it also makes the spatial structure of TM different. Therefore, TM prediction has become one of the research hotspots [36]. A large body of research has focused on TM forecasting [21], including using a combination and integration of forecasts, using nonlinear methods for forecasting, and extending existing methods to better model the changing nature of tourism data [37]. The gravity model is an earlier method used to analyze international TM [38]. Due to its effectiveness in explaining TM [22], gravity models are often used to analyze international tourism service trade. Although the use of gravity models to predict bilateral TM still lacks a corresponding theoretical explanation mechanism, empirical evidence supports the applicability and robustness of gravity models for TM [23]. Existing research focuses on examining the movement patterns and spatial structure of international TM in destinations [39], such as the transfer of inbound TM within regions and the influencing factors of inbound TM within destinations [40]. There are still few studies on the overall spatial characteristics of TM within destination countries, and the only literature is mainly based on digital footprints or questionnaire data to analyze the spatial structure of TM [41, 42].

Unlike the tourist arrivals indicator, which focuses more on the mobility of people, TM examines a wider range of content, including the mobility of people, the mobility of materials, the mobility of ideas (more intangible thoughts and fantasies), and the mobility of technology [8]. The early tourist movement focused more on tourist travel decisions and the resulting movement patterns. Lue et al. [43] summarized five travel patterns of tourists between destinations. Li et al. [44] revealed the spatial patterns of TM and tourism propensity in the Asia-Pacific region over the past 10 years. McKercher and Lau [45] took Hong Kong as an example and identified 78 movement patterns and 11 movement styles of TM within the destination. In recent years, with the help of technologies such as GPS, GIS, and RFID, the movement of tourists within scenic spots has attracted attention [46]. Research on visitor movement in national parks, theme parks, protected areas, etc. continues to increase [47–49], and explore the influencing factors of visitor movement [50], broadening the microscale visitor mobility research

content. TM also has economic, social, and cultural impacts on destinations through the movement of tourists. Numerous empirical studies have shown that tourist arrivals have a positive impact on economic growth [51]. Tourism is an important driver of economic growth [52]. However, some studies have shown that tourist arrivals do not directly lead to economic growth, but promote TEG through regional economic development [53–55]. The mobility of tourism will also bring about changes in destination transportation facilities. Transportation is not only an important carrier of TM but also an important part of tourists' travel experience [8]. It also has a positive impact on destination company value together with TM [26].

There are many theoretical discussions and empirical studies on the factors influencing TEG. From the perspective of suppliers, resource endowment [56–58] and environmental quality [59–62] are the fundamental factors determining tourism development. Simultaneously, as a typical service industry, human capital and physical capital in the tourism industry [63, 64] and service level [65] will impact tourism economic efficiency. From the perspective of demanders, the rise of per capita income and consumption upgrading continue to drive the transformation in the tourism industry [66], which in turn leads to an increasing scale of market demand [67], which provides the possibility of increasing the foreign exchange earnings, local capital accumulation, and consumption spillovers. From the perspective of supporters, scholars have verified the significant effects of factors on TEG, including the transportation facilities and accessibility [68–71], the basis of the economy and marketization [72], industrial structure [73], public policy [74–76], and technological progress [77].

In summary, the research on TM has paid attention to its impact on the regional economy, but they both ignored the role of TM on TEG. Studies of TEG based on static factors have primarily relied on econometric models [78]. Although the spatial spillover effects of influencing factors have gradually gained attention, its depth is limited and fails to explore the impact of TM and other related factors on the TEG. TM is becoming central to tourism activities and understanding the capital mobility of tourism will have implications for tourism development under the new mobility paradigm [79]. This study proposes the concept of TM based on the theory of fluid mechanics, explores its impact on TEG, and analyzes the contribution of each influencing factor to TEG.

## Theoretical framework, research methods, and data sources

### Theoretical framework

Traditionally, tourism research considers the tourism system as tourist sources, tourist destinations, and tourist corridors (transportation systems) [80, 81]. Under the new mobility paradigm, this study regards the spatial transfer of tourists from the source to the destination as a mobility process. Tourist mobility is the fundamental reason for the existence of tourism. If tourists stop flowing, tourism will cease to exist.

It is known that the fluid will be affected by a variety of factors, such as viscosity, density, resistance coefficient, and altitude. As shown in Fig 1, the total mobility of tourists from a tourist origin to a tourist destination is the number of tourists (Q). The spatial transfer of tourists, on the other hand, requires the use of transportation infrastructure as well as means of delivery. As an essential vehicle to support tourism development, transportation infrastructure directly reflects regional accessibility and relevance and is a crucial factor influencing TM [82–84], and its construction level has different effects on TEG in different regions [11, 85–87]. According to the equations in fluid mechanics, the average velocity is equal to the flow rate ratio to the cross-sectional area. It can be deduced that TM = Q/TL. TM is determined by the number of tourists (Q) and the length of transportation infrastructure (TL). According to the definition, this indicator considers both tourist arrivals and flow rate, and its significance lies

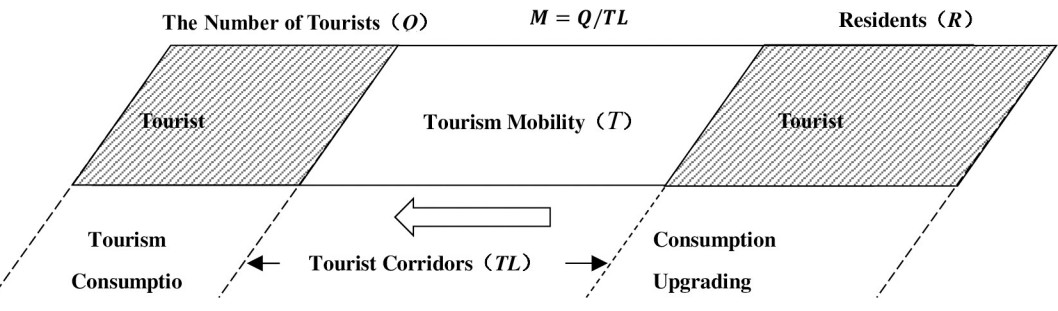

**Fig 1. The influence of TM on TEG.**

in its ability to characterize the mobility of tourism factors relying on tourists and physical transportation. This paper also connects the factor decomposition method to determine the importance of TM to TEG and presents theoretical implications for identifying essential factors to enhance tourism efficiency and stimulate tourism industry development.

## Research methods

**Measurement of tourism mobility.**　The basic principle of fluid mechanics is that the average velocity is proportional to the flow rate and inversely proportional to the cross-sectional area. This paper characterizes the flow rate by the number of tourist inflows, and the length of transportation infrastructure represents the cross-sectional area, with the equation as (1)

$$TM_i^t = Q_i^t / TL_i^t \tag{1}$$

where $TM_i^t$ represents the TM in province i and year t; $Q_i^t$ is the number of tourists in province i and year t; and $TL_i^t$ is the length of the weighted transportation infrastructure in province i and year t, including railroads, highways, primary roads, secondary roads, and other grades of roads. China's railway and road passenger traffic accounts for the vast majority of the total passenger traffic. Furthermore, we were unable to calculate weighted air and water transportation infrastructure lengths, so we only consider the land transportation infrastructure data. The length of transportation infrastructure is weighted according to Chen et al. [88].

**Exploratory spatial data analysis.**　It is generally believed that tourism has a spatial spillover effect and spatial correlation [28]. Therefore, we use Exploratory Spatial Data Analysis (ESDA) to detect spatial correlation among the variables. ESDA is used to analyze spatial characteristics through global and local spatial autocorrelation measurements [42, 89].

The global Moran's I is an indicator of whether factors are spatially correlated and its value ranges from -1 to 1. When $0 < I \leq 1$, it indicates a positive spatial correlation; when $-1 \leq I < 0$, it indicates a negative spatial correlation; when I = 0, there is no spatial relationship. The equation is as in (2).

$$I = \frac{n \sum_{i=1}^{n} \sum_{j=1}^{n} W_{ij} |TEG_i - \overline{TEG}| |TR_j - \overline{TEG}|}{\sum_{i=1}^{n} \sum_{j=1}^{n} W_{ij} \sum_{i=1}^{n} |TR_i - \overline{TEG}|^2} \tag{2}$$

Where $TEG_i$ and $TEG_j$ denote the tourism revenue of provinces i and j, respectively; n is the number of provinces; $\overline{TEG}$ denotes the average value of tourism revenue of each province; $W_{ij}$ represents the spatial weight matrix of provinces i and j. We choose the adjacency matrix and use Guangdong and Guangxi as the neighboring provinces of Hainan.

Local spatial autocorrelation is used to explore cluster patterns and spatial patterns [90, 91]. We analyze the local spatial autocorrelation characteristics through cluster and outlier analyses. The calculation process is expressed as Formula (3):

$$I_p = \frac{(n-1)\sum_{q=1,q \neq p}^{n} W_{pq}(TEG_p - \overline{TEG})(TEG_q - \overline{TEG})}{\sum_{q=1,q \neq p}^{n}(TEG_q - \overline{TEG})^2} \tag{3}$$

With a Z statistical test as in Formula (4), the cluster and outlier analyses can identify H_H (High_High) clusters, L_L (Low_Low) clusters, L_H (low value surrounded by high values) clusters, and H_L (high value surrounded by low values) clusters at a 95% confidence level.

$$Z_{I_p} = \frac{I_p - E[I_p]}{\sqrt{V[I_p]}} \tag{4}$$

**Econometric model.** The econometric model, including tourism economic growth (TEG), tourism mobility (TM), physical capital in the tourism industry (TP), and human capital in the tourism industry (TH), is constructed according to economic growth theory without considering spatial spillover effects. Besides, since the measurement of TM only considers land transportation infrastructure data, the passenger traffic by the airport (TA) is introduced in the model to characterize the air capacity. Eq (5) represents the econometric model ($TEG_{it}$) in province i and year t, where $\alpha$ is the constant term, $\beta$ is the parameter to be estimated, $\mu_i$ denotes the spatial effect, and $\varepsilon_{it}$ denotes the random error term.

$$Ln(TR)_{it} = \alpha + \beta Ln(TM)_{it} + \beta_1 Ln(TP)_{it} + \beta_2 Ln(TH)_{it} + \beta_3 Ln(TA)_{it} + \mu_i + \varepsilon_{it} \tag{5}$$

However, the spatial correlation of TEG will lead to biased parameter estimates of traditional econometric models. If the test results of global Moran's I indicate that TEG is significantly spatially correlated, a spatial econometric model should be introduced to solve the bias-variance problem. The spatial Durbin model (Eq 6) is developed according to Eq 5. The spatial weight matrix used in the spatial Durbin model is an adjacency matrix. $y_{it}$ represents the TEG in province i and year t; $x_{it}$ represents the TM, TP, TH, and TA in province i and year t; and $W_{ij}y_{jt}$ and $W_{ij}x_{jt}$ are the TEG and lagged terms of each influencing factor, respectively. $\rho$ and $\varphi$ are spatial lagging coefficients, and $v_t$ denotes the time effect.

$$y_{it} = \rho \sum_{j=1}^{n} W_{ij}y_{jt} + \beta x_{it} + \sum_{j=1}^{n} \varphi W_{ij}x_{jt} + \mu_i + v_t + \varepsilon_{it} \tag{6}$$

**LMDI decomposition.** The LMDI decomposition method is widely used because it can effectively solve the residual problem in the decomposition and zero and negative values in the data. LMDI In this study, TEG is decomposed according to Eq (7). The influencing factors of TEG are decomposed into tourism mobility effects (*TE*), cumulative traffic effects (*Traffic*), effects of the tertiary industry (*Industry*), structural effects of the tourism industry (*Structure*), and reception effects (*Reception*). The equations are shown in (8) to (11). Traffic indicates the weighted road length; GDP (service) intimates the value added of the tertiary industry; Population represents the population in each province, and Visitors is the number of tourists. Introducing the log-average function L(x,y) defined in Eq (12). Eq (7) is decomposed into Eq (13) by LMDI, where ΔTEG denotes the amount of change in TEG from initial time 0 to period t, and ΔTM, ΔT, ΔI, ΔS, ΔW represent the contribution of each influencing factor to TEG. The

equations are shown in (14) to (18).

$$TEG = TE * Traffic * Industry * Structure * Reception \qquad (7)$$

$$TM = Visitors/Traffic \qquad (8)$$

$$Industry = GDP(service)/Population \qquad (9)$$

$$Structure = TEG/GDP(service) \qquad (10)$$

$$Reception = Population/Visitors \qquad (11)$$

$$L(x, y) = \begin{cases} \dfrac{x - y}{\ln x - \ln y}, x \neq y \\ x, x = y \\ 0, x = y = 0 \end{cases} \qquad (12)$$

$$\Delta TEG = \Delta TEG(t) - \Delta TEG(0) = \Delta TM + \Delta T + \Delta I + \Delta S + \Delta W \qquad (13)$$

$$\Delta TM = \frac{\Delta TEG(t) - \Delta TEG(0)}{\ln TEG(t) - \ln TEG(0)} * \ln\left(\frac{TM(t)}{TM(0)}\right) \qquad (14)$$

$$\Delta T = \frac{\Delta TEG(t) - \Delta TEG(0)}{\ln TEG(t) - \ln TEG(0)} * \ln\left(\frac{Traffic(t)}{Traffic(0)}\right) \qquad (15)$$

$$\Delta I = \frac{\Delta TEG(t) - \Delta TEG(0)}{\ln TEG(t) - \ln TEG(0)} * \ln\left(\frac{Industry(t)}{Industry(0)}\right) \qquad (16)$$

$$\Delta S = \frac{\Delta TEG(t) - \Delta TEG(0)}{\ln TEG(t) - \ln TEG(0)} * \ln\left(\frac{Structure(t)}{Structure(0)}\right) \qquad (17)$$

$$\Delta W = \frac{\Delta TEG(t) - \Delta TEG(0)}{\ln TEG(t) - \ln TEG(0)} * \ln\left(\frac{Reception(t)}{Reception(0)}\right) \qquad (18)$$

## Data sources

The study area is 31 provinces of China (excluding Hong Kong, Macao, and Taiwan), which is divided into seven regions according to the geographical divisions of China. The provinces included in each region are listed in supporting information. Since data availability varies widely across regions, the research period of TM and LMDI decomposition is from 2000 to 2018. As the National Bureau of Statistics of China (NBS) started to collect the employment data of private enterprises and individuals by sector in 2004 and the data for 2018 has not been updated yet, the research period of the spatial econometric model only covers the period from 2004 to 2017.

The data sources involved in the paper are as follows: the transportation infrastructure data come from the China Statistical Yearbook; the number of tourists is obtained from the Statistical Bulletin on National Economic and Social Development. Air passenger traffic data is collected from Civil Aviation Airport Production Statistics Bulletin. We employ the social fixed asset investment in transportation, storage, and postal services, wholesale and retail trade,

accommodation and catering, and culture, sports, and entertainment as proxies for physical capital in the tourism industry (TP). This is because various aspects influence tourism development. Considering that only direct tourism investment does not reflect the total investment in tourism by society, we choose the four industries closely related to tourism development as physical capital in the tourism industry.

In this paper, private and individual employees in the transport, storage, and postal industry, wholesale and retail trade, and accommodation and catering industries are used to represent the human capital in the tourism industry (TH). The main reason for this is that, on the one hand, most studies only consider the number of employees in travel agencies, scenic spots, and star hotels, which differs significantly from the actual number of direct and indirect employees in tourism. On the other hand, since private enterprises and individual employment solve more than 80% of the urban employment problem, the number of private enterprises and individual employment in the three industries related to the tourism industry is chosen to represent the human capital. All the above data are collected from the NBS (http://data.stats.gov.cn). In the LMDI decomposition, the value added of the tertiary industry and the population in each province come from the China Statistical Yearbook.

## Analysis of tourism mobility measurement results

### Spatiotemporal evolution characteristics of tourism mobility

Limited by space, Table 1 only shows the results of TM over five years. During the study period, TM increased from 56~12745 p visitors /km to 382~18865 p visitors /km, with an average annual growth rate between 2.20% and 13.46%. According to the average value of TM (Fig 2), the study areas are divided into the following three types.

1. "Leading Area", including East China and North China, ranked first and second in all regions. Their TM increased from 2679.39 and 1884.34 p visitors/km in 2000 to 5859.93 and 5209.94 p visitors/km in 2018. However, their annual average growth rates were 5.07% and 6.43%, respectively, ranking first and second from the bottom in all regions. East China is located on the coast, relying on superior natural conditions and an economic foundation, and its regional transportation system is relatively complete. Therefore, it has formed many advantageous tourist resource gathering areas and has become the main tourist destination of inbound tourists in China, and its mobility has long ranked first in the country. As a political and economic center, Beijing has become a tourist attraction for domestic and inbound tourism with a large number of historical and cultural tourism resources. It also drives the joint development of the tourism industry in North China with the Beijing-Tianjin-Hebei urban agglomeration as the core, making North China the second largest core area of TM after East China.

2. "Stable Area", including South China, Southwest China, Central China, and Northeast China, ranked third to sixth in all regions. Their TM increased from 903.57p visitors/km, 695.15p visitors/km, 632.06p visitors/km, 493.33 p visitors/km in 2000 to 2626.11p visitors/km, 2754.97p visitors/km, 2857.88p visitors/km, 2244.68 p visitors/km in 2018. The average annual growth rates were 6.58%, 8.81%, 9.06%, and 9.38%, respectively. TM in South China grew rapidly during 2005~2015, while it has gradually slowed down in recent years. This is mainly due to the construction of the early transportation system in South China, which increased tourist mobility. After the basic construction of facilities, the incremental tourist inflows decreased, and the overall growth remained stable. Central China has become one of the core transportation hubs under its location and has driven regional tourism development, becoming a central province in the second echelon of TM. Due to geographical

**Table 1. Evaluation of TM (unit: p visitor/km).**

| Provinces | 2000 | 2005 | 2010 | 2015 | 2018 |
|---|---|---|---|---|---|
| Beijing | 5815.61 | 6244.13 | 6396.40 | 8802.59 | 9749.22 |
| Tianjin | 2423.53 | 3244.66 | 2806.19 | 7044.53 | 9087.30 |
| Hebei | 643.89 | 861.74 | 808.73 | 1635.52 | 2808.76 |
| Shanxi | 442.66 | 751.96 | 848.94 | 2084.30 | 3911.71 |
| Inner Mongolia | 95.99 | 221.08 | 248.93 | 380.48 | 492.70 |
| Liaoning | 686.93 | 1284.09 | 2478.02 | 2269.75 | 4008.91 |
| Jilin | 385.53 | 441.30 | 627.62 | 1202.56 | 1710.53 |
| Heilongjiang | 407.54 | 532.26 | 1093.06 | 760.01 | 1014.61 |
| Shanghai | 12745.08 | 8117.20 | 13246.92 | 15309.97 | 18864.86 |
| Jiangsu | 1992.54 | 1819.48 | 2047.88 | 3226.98 | 4053.52 |
| Zhejiang | 1347.43 | 2177.08 | 2358.99 | 3762.31 | 4609.22 |
| Anhui | 573.09 | 576.14 | 951.49 | 2089.90 | 3015.35 |
| Fujian | 639.86 | 1001.07 | 1339.40 | 2378.62 | 3825.36 |
| Jiangxi | 703.39 | 866.53 | 879.65 | 2419.10 | 4047.48 |
| Shandong | 754.32 | 1269.68 | 1317.57 | 2116.74 | 2603.69 |
| Henan | 701.00 | 974.92 | 1182.94 | 2121.21 | 2715.03 |
| Hubei | 922.82 | 807.60 | 977.49 | 1811.08 | 2295.51 |
| Hunan | 1086.90 | 1183.50 | 1001.53 | 1922.76 | 2876.79 |
| Guangdong | 694.74 | 863.53 | 1941.69 | 3419.35 | 1874.52 |
| Guangxi | 744.77 | 1013.08 | 1404.58 | 2434.53 | 4532.18 |
| Hainan | 645.95 | 987.93 | 1007.22 | 1648.52 | 1858.19 |
| Chongqing | 1214.23 | 1552.96 | 1798.32 | 3056.53 | 3919.89 |
| Sichuan | 654.32 | 1346.55 | 1176.15 | 1966.18 | 2027.73 |
| Guizhou | 878.41 | 710.70 | 1540.86 | 2521.01 | 5206.88 |
| Yunnan | 357.47 | 567.94 | 804.54 | 1465.18 | 2753.10 |
| Tibet | 55.86 | 159.35 | 178.99 | 326.26 | 381.81 |
| Shaanxi | 671.95 | 944.96 | 918.24 | 2108.71 | 3256.66 |
| Gansu | 198.24 | 279.20 | 428.90 | 1089.77 | 1925.55 |
| Qinghai | 165.60 | 194.06 | 216.32 | 284.30 | 474.31 |
| Ningxia | 191.20 | 277.46 | 349.91 | 416.67 | 704.38 |
| Xinjiang | 183.04 | 175.42 | 253.72 | 354.96 | 777.00 |

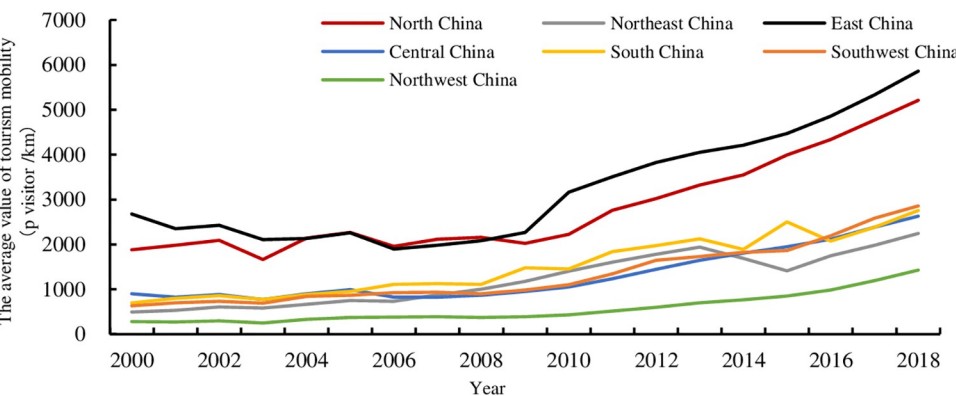

**Fig 2. Mean value of regional TM from 2000 to 2018.**

restrictions, Northeast and Southwest China are less connected to the transportation network than coastal areas, resulting in relatively low levels of TM. Northeast China focuses on the development of heavy industry but pays little attention to the tertiary industry, and tourism infrastructure construction and resource development are relatively weak, which leads to low TM. There are many mountains in Southwest China, and its early traffic development level lags. With the opening of the Chengdu-Chongqing high-speed railway and Chengdu-Guizhou high-speed railway, and the development of the air transportation industry, the land and air transportation layout in Southwest China is becoming increasingly mature. Southwest China actively developed its resources, and the tourist inflow increased from 145 million (2000) to 2.994 billion (2018), with an average value of TM catching up with that of southern China during 2016~2018.

3. "Potential Area", including Northwest China, ranks last in terms of average tourist mobility. Its TM increased from 282.01 p visitors/km in 2000 to 1427.58 p visitors/km in 2018, but its average annual growth rate was 10.01%, ranking first among all regions. As less developed region, Northwest China has a poor foundation in economic development and openness to the outside world, and TM has long been at the bottom of the list. Although TM in Northwest China has long been at the bottom of the list, its mobility growth rate leads other regions as tourism infrastructure construction and resource development levels have improved under the active promotion of Western Development policies, the Five-Year Plan, and the Territorial Tourism Strategy.

To more intuitively observe the temporal and spatial change characteristics of TM during the study period, we apply the method of natural breaks to classify the 31 provinces. Natural breaks classes are based on natural groupings inherent in the data. Class breaks are identified that best group similar values and maximize the differences between classes. The features are divided into classes whose boundaries are set where there are relatively big differences in the data values. The natural breaks classification method is a data classification method designed to determine the best arrangement of values into different classes. This is done by seeking to minimize each class's average deviation from the class mean while maximizing each class's deviation from the means of the other groups [92]. We divided the 31 provinces into five categories, highest-value area, higher-value area, medium-value area, lower-value area, and lowest-value area, according to the TM in 2000, 2005, 2010, 2015, and 2018. As shown in Fig 3, (1) Shanghai and Beijing have long been in the highest-value area and higher-value area of TM. Tibet, Qinghai, Ningxia, Xinjiang, Inner Mongolia, Gansu, Jilin, Heilongjiang, Hubei, and Hainan have long been in the lowest-value and lower-value areas. (2) Over time, the number of provinces in the highest-value area and the higher-value area increased significantly, from 2 provinces in 2000 to 12 provinces in 2018. The number of provinces in the lowest-value area and lower-value area significantly decreased, from 26 provinces in 2000 to 12 provinces in 2018; the number of provinces in the medium-value area fluctuated randomly, with the fewest 3 in 2000 and the most 13 in 2015. (3) Except for Shanxi, Northwest China has been in the lowest-value area and the lower-value area for a long time; The TM values in Southwest China have changed greatly. Chongqing and Guizhou have jumped from the lower-value area to the higher-value area, and Yunnan has jumped from the low-value area to the medium-value area. Tibet is relatively stable and has been in the lowest-value area for a long time; South China is relatively stable, but the average value TM in Guangxi has changed greatly, jumping from the lower-value area to the higher-value area; The average TM in Central China has been in the low-value area for a long time. Central China is also relatively stable, and its average TM has long been located in the lower-value area and the medium-value area. Except for Shanghai, which has always been in the highest-value area, the initial value of TM in other provinces in

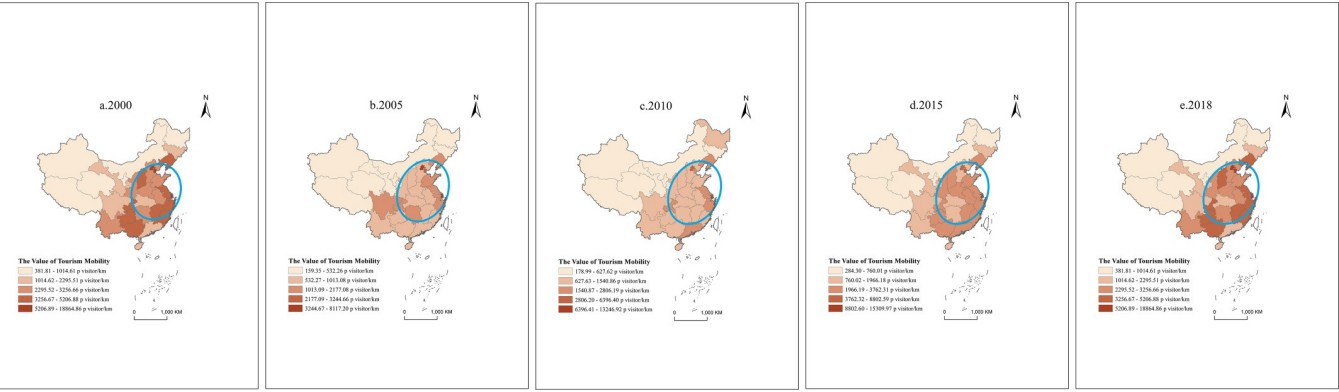

**Fig 3. The spatiotemporal pattern and direction distribution of provincial TM.** a. 2000, b. 2005, c. 2010, d. 2015, e. 2018.

East China has jumped upward. In the Northeast, Liaoning's TM has always been in a leading position, and it has gradually transitioned from a lower-value area to a higher-value area. However, Jilin and Heilongjiang have always been in the lowest-value area and the lower-value area, respectively. Changes in TM in North China are diverse. Beijing has long been located in the highest-value area and higher value area. Inner Mongolia has been in the lowest-value area for a long time. Hebei is in the lower-value area most of the time. Tianjin and Shanxi changed greatly and finally jumped to the highest-value area and the higher-value area, respectively.

We use the standard deviation ellipse to identify the direction of TM in each province. As shown in Fig 3, the lengths of the minor semiaxis and major semiaxis of the ellipse increased significantly. The growth of the short semiaxis reveals that the degree of dispersion of TM in China's provinces is gradually increasing. This result is consistent with the previous analysis conclusions that TM in some provinces shows a more obvious transition trend, which makes the overall dispersion of TM increase.

### Spatial correlation characteristics of tourism mobility

**Global spatial autocorrelation of tourism mobility.** We use ArcGIS 10.8 to calculate the global Moran's I of TM for 2000–2018, and the results are shown in the table below (Table 2). The global Moran's I values from 2000 to 2018 were all positive, and the results from 2000 to 2015 were not significant, and the results from 2016 to 2018 were all significant at the 90% level. TM presents a significant positive spatial correlation. This shows that provinces with high TM in China have relatively high TM in their surrounding areas. From the overall trend,

**Table 2. Global Moran's I index of TM.**

| Year | Moran's *I* | Z- value | P-value | Year | Moran's *I* | Z -value | P-value |
|------|-------------|----------|---------|------|-------------|----------|---------|
| 2000 | 0.015 | 0.876 | 0.381 | 2010 | 0.005 | 0.662 | 0.508 |
| 2001 | 0.009 | 0.710 | 0.478 | 2011 | 0.021 | 0.889 | 0.374 |
| 2002 | 0.024 | 0.948 | 0.343 | 2012 | 0.021 | 0.878 | 0.380 |
| 2003 | 0.035 | 1.139 | 0.255 | 2013 | 0.027 | 0.968 | 0.333 |
| 2004 | 0.062 | 1.430 | 0.153 | 2014 | 0.058 | 1.413 | 0.158 |
| 2005 | 0.075 | 1.615 | 0.106 | 2015 | 0.071 | 1.601 | 0.109 |
| 2006 | 0.026 | 0.874 | 0.382 | 2016 | 0.083 | 1.783 | 0.075* |
| 2007 | 0.052 | 1.239 | 0.215 | 2017 | 0.081 | 1.745 | 0.081* |
| 2008 | 0.063 | 1.409 | 0.159 | 2018 | 0.076 | 1.671 | 0.095* |
| 2009 | 0.020 | 0.796 | 0.426 | | | | |

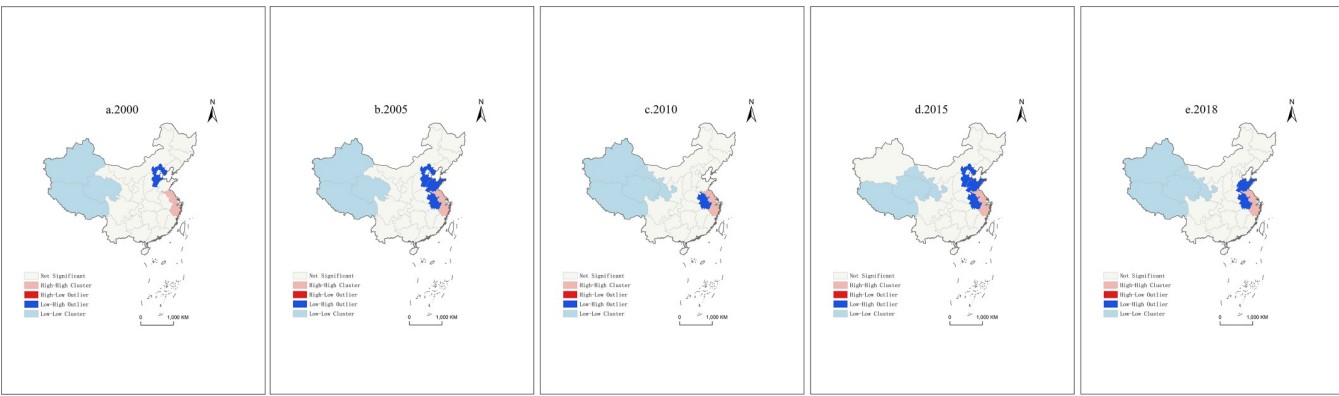

**Fig 4. LISA clustering results of TM.** a. 2000, b. 2005, c. 2010, d. 2015, e. 2018.

the spatial correlation degree of China's TM has gradually increased, but its value has not exceeded 0.1, indicating that the spatial agglomeration effect of China's TM is still weak.

**Local spatial autocorrelation cluster of tourism mobility.** The global Moran's I cannot reflect the spatial correlation exhibited by local regions or individual provinces. We further use ArcGIS 10.8 to draw the LISA cluster diagram for 2000, 2005, 2010, 2015, and 2018 (Fig 4). The research samples are divided into four types of agglomeration: provinces with high TM are surrounded by provinces with high TM (H-H agglomeration), provinces with high TM are surrounded by provinces with low TM (H-L agglomeration), provinces with low TM are surrounded by provinces with high TM (L-H agglomeration), and provinces with low TM are surrounded by provinces with low TM (L-L agglomeration).

The results show that (1) provinces with H-H aggregation of TM in different periods are relatively stable; L-L and L-H aggregation types are stable but mixed with changes; The H-L aggregation type does not appear, which indicates that there is no "darkness under the light" area for China's provincial TM. Provinces with high TM can improve the TM of weekly provinces to a certain extent. (2) The H-H agglomeration is mainly concentrated in Jiangsu and Zhejiang. These regions are economically developed and have high per capita discretionary income. Moreover, the tourism infrastructure in these regions is more complete than that in other regions, and the tourist reception scale is also higher, so their TM shows a high local concentration. (3) The L-L agglomeration types are mainly distributed in geographically remote areas such as Qinghai, Tibet, Gansu, and Xinjiang in inland China. Moreover, Xinjiang and Gansu temporarily withdraw from the L-L agglomeration area. The main reason for this pattern is that the transportation infrastructure in the areas above mentioned is relatively underdeveloped. The "space-time compression effect" brought about by the rapid development of China's transportation is not significant. Furthermore, due to the distance from the main tourist source markets, although the TM shows a high growth rate, it is still in the lowest-value area and the lower-value area for a long time. (4) L-H agglomeration is mainly transferred in Anhui, Shandong and Hebei, and these provinces are located in the "Leading Area". The average value of TM in the surrounding provinces is generally high, forming a "collapse area" for TM.

## The impact of tourism mobility on tourism economic growth

### Spatial autocorrelation of tourism economic growth

In this study, a Monte Carlo simulation was selected to analyze the spatial autocorrelation of TEG (Table 3). Moran's I was positive from 2000 to 2018. They passed the significance test of different

**Table 3. Moran's I index of TEG.**

| Year | Moran's I | Z-Score | P-Score | Year | Moran's I | Z-Score | P-Score |
|------|-----------|---------|---------|------|-----------|---------|---------|
| 2000 | 0.117 | 1.405 | 0.086* | 2010 | 0.317 | 3.177 | 0.005*** |
| 2001 | 0.113 | 1.406 | 0.087* | 2011 | 0.307 | 2.989 | 0.007*** |
| 2002 | 0.110 | 1.233 | 0.083* | 2012 | 0.241 | 2.401 | 0.009*** |
| 2003 | 0.109 | 1.428 | 0.080* | 2013 | 0.215 | 2.312 | 0.017** |
| 2004 | 0.117 | 1.335 | 0.086* | 2014 | 0.234 | 2.273 | 0.006*** |
| 2005 | 0.118 | 1.455 | 0.082* | 2015 | 0.249 | 2.535 | 0.008*** |
| 2006 | 0.105 | 1.188 | 0.101 | 2016 | 0.262 | 2.666 | 0.007*** |
| 2007 | 0.101 | 1.210 | 0.100* | 2017 | 0.324 | 3.147 | 0.002*** |
| 2008 | 0.111 | 1.302 | 0.096* | 2018 | 0.349 | 3.506 | 0.001*** |
| 2009 | 0.281 | 2.793 | 0.008*** | | | | |

Note

***, **, * indicate passing the significance test at the 1%, 5%, and 10% levels, respectively.

degrees except in 2006, indicating that TEG has a significant positive spatial correlation. Therefore, a spatial econometric model should be selected to analyze the influencing factors of TEG.

## Traditional panel data model

The unit root test using LLC and Fisher showed no unit root for TEG, TM, TH, TP, and TA (Table 4). The Kao test, Pedroni test, and Westerlund test were used to determine the cointegration relationship between the variables. The test results showed a cointegration relationship, indicating that the data can be used for modeling.

In terms of the regression model, the BP Lagrangian test results show the rejection of the mixed model. Wooldridge and Wald's test indicates the presence of heteroskedasticity and autocorrelation in the data. The presence of heteroskedasticity would lead to an increase in the variance of the model parameters and invalidate the Hausman test results. If the regression is still performed using the method without heteroskedasticity, it will undermine the validity of the t-test and F-test, while autocorrelation will exaggerate the significance of the parameters. Therefore, the panel model is selected by the over-identification test (Hausman test result is significant), and the result shows that the Sargan-Hansen statistic is 14.32 and significant, so fixed effect modeling should be selected.

**Table 4. Results of stationarity test.**

| Unit root test | TEG | TM | TP | TH | TA |
|----------------|-----|-----|-----|-----|-----|
| LLC test | -4.6610*** | -19.0978*** | -4.1181*** | -6.8463*** | -9.0320*** |
| Fisher test | 139.3618*** | 86.8109** | 88.8039** | 98.7588*** | 92.5520*** |
| **Cointegration tests** | | | | | |

| Kao test | | Pedron test | | Westerlund test | |
|----------|-----|-------------|-----|-----------------|-----|
| Modified Dickey-Fuller t | -2.0376** | Modified Phillips-Perron t | 5.6168** | Variance ratio | 2.6208*** |
| Dickey-Fuller t | -3.5593*** | Phillips-Perron t | -7.6942*** | | |
| Augmented Dickey-Fuller t | -2.2142** | | | | |
| Unadjusted modified Dickey-Fuller t | -2.6291*** | Augmented Dickey-Fuller t | -4.7115*** | | |
| Unadjusted Dickey-Fuller t | -3.8579*** | | | | |

Note

***, **, * indicate passing the significance test at the 1%, 5%, and 10% levels, respectively.

**Table 5. Regression test results.**

| Variables | Fixed effects model | | Random effects model | |
|---|---|---|---|---|
| | Coefficients | Driscoll-Kraay standard errors | Coefficients | Driscoll-Kraay standard errors |
| *LnTM* | 0.6186*** | 0.0341 | 0.4525 | 0.3299 |
| *LnTP* | 0.2341*** | 0.0308 | 0.5849*** | 0.0370 |
| *LnTH* | 0.3276*** | 0.0526 | 0.4335*** | 0.0625 |
| *LnTA* | 0.5250*** | 0.0197 | 0.0102 | 0.0230 |
| *_cons* | −9.3119*** | 0.1385 | −3.0771** | 1.2723 |
| R-squared | 0.9441 | | 0.9234 | |
| F-score | 6982.81*** | | Wald test | 5040.52*** |
| F-test | 25.25*** | | Hausman Test | 26.68*** |
| Wooldridge test | 5.692** | | Wald test | 19391.36*** |
| Sargan–Hansen test | 14.32*** | | BP Lagrange multiplier test | 433.09*** |

Note

***, **, * indicate significance at the 1%, 5%, and 10% levels, respectively.

To further address heteroskedasticity and autocorrelation, this study uses Driscoll-Kraay standard errors for regression. The results in Table 5 show a significant positive effect of each variable on TEG, where each 1% increase in TM will promote 0.62% growth in the tourism economy.

## Spatial panel data model

In this paper, the specific form of the spatial panel data model was determined by LM-LAG and LM-ERROR tests. If the result of LM-lag is significant and LM-error is not significant, then SLM should be used, and vice versa, SEM should be used. If LM-lag and LM-error statistics are significant, it indicates that the spatial correlation of the lag term and the spatial correlation of the residuals should be considered. In this case, the SDM can be used to set the model. Subsequently, this study determined whether the SDM model would degenerate into SLM or SEM by Wald and LR tests, and the results showed that all passed the significance test. Meanwhile, the test results of LM-lag, LM-error, LM-lag (robust), and LM-error (robust) were significant (Table 6), indicating that the model set using SDM has a certain rationality.

We selected the regression model through the Hausman test, and the result showed that the value was 19.31, and the corresponding probability value was 0.007, which indicated that the null hypothesis of random effect was rejected. Therefore, the fixed-effect model was selected for regression analysis. Table 7 shows the estimation results, where ρ rejects the original hypothesis only in the Spatio-temporal fixed-effects model. Therefore, this paper provides a

**Table 6. Tests of the spatial panel data model.**

| Statistical quantities | Value | Statistical quantities | Value |
|---|---|---|---|
| LM-lag | 1355.52*** | LR-lag | 42.17*** |
| LM-error | 3519.48*** | LR-error | 126.61*** |
| LM-lag (robust) | 6294.22*** | Wald-lag | 42.58*** |
| LM-error (robust) | 8458.17*** | Wald-error | 48.83** |

Note

***, **indicate passing the significance test at the 1% and 5% levels, respectively.

**Table 7. Estimation results of SDM.**

| Variables | Spatial fixed effects | Temporal fixed effects | Spatio-temporal fixed effects |
|---|---|---|---|
| *LnMobility* | 0.512*** | 0.352*** | 0.482*** |
| *LnTP* | −0.020 | 0.635*** | −0.032 |
| *LnTH* | 0.114 | 0.348*** | 0.140** |
| *LnTA* | 0.263*** | 0.055* | 0.264** |
| Wx *LnTM* | −0.400** | 0.843*** | −0.449 |
| Wx *LnTP* | 0.506* | 0.286 | 0.993*** |
| Wx *LnTH* | 1.081*** | 0.577** | 1.693*** |
| Wx *LnTA* | −0.129 | −0.655*** | −0.215 |
| *R²* | 0.9604 | 0.900 | 0.9602 |
| *Spatial rho* | −0.102 | −0.641 | −0.559*** |
| *Variance sigma2_e* | 0.030*** | 0.100*** | 0.031*** |

Note

***, **, * indicate significance at the 1%, 5%, and 10% levels, respectively.

specific analysis of the Spatio-temporal fixed-effects model. The regression results indicate that TM shows a significant positive effect on regional TEG.

According to the results and spatial effect decomposition (Table 8), ρ is -0.559, indicating that the growth of the tourism economy in neighboring provinces will have a negative impact on the local area. The direct effect of TM is significant, indicating that TM will promote TEG. However, the indirect effect results show that the increase in TM in neighboring provinces will have a negative impact on the local TEG.

**Decomposition of the influencing factors by LMDI.** We decompose the influencing factors and analyze their contribution trend. Table 9 shows the specific contribution of each influencing factor to the TEG in the seven regions.

ΔT increases from 15.41% in 2000~2005 to 22.55% (2005~2010), and then decreases to 9.35% in 2010~2015 and 7.01% in 2015~2018. Overall, the ΔT showed a downward trend, but it is still an important factor in promoting TEG. The average contribution rate of the ΔT from 2000 to 2018 reached 14.82%.

ΔI maintained an overall downward trend during 2000 ~2018. It gradually decreased from 31.42% (2000~2005) to 22.94% (2015~2018). In contrast, the added-value of tertiary industry per capita increases from 3653 yuan to 34,969 yuan in the same period, indicating that the contribution of tertiary industry to TEG continues to decline, and tourism is gradually decoupled from the development of the tertiary industry.

ΔS maintained an overall upward trend during 2000~2018, from 7.09% (2000~2005) to 14.67% (2015~2018). The overall contribution rate was 11.50%, indicating that increasing the proportion of the tertiary industry in tourism can promote TEG.

**Table 8. Decomposed spatial effects of SDM.**

| Variables | Direct effect | Indirect effect | Total effect |
|---|---|---|---|
| *LnTM* | 0.497*** | −0.486* | 0.011 |
| *LnTP* | −0.051 | 0.685*** | 0.634*** |
| *LnTH* | 0.104* | 1.100*** | 1.204*** |
| *LnTA* | 0.275*** | −0.241 | 0.035 |

Note

***, * indicate passing the significance test at the 1% and 10% levels, respectively.

**Table 9. Contribution rate of factors influencing regional TEG.**

| Time period | Factors | North China | Northeast China | East China | Central China | South China | Southwest China | Northwest China | Nationwide |
|---|---|---|---|---|---|---|---|---|---|
| 2000~2005 | $\Delta TM$ | 15.76% | 14.84% | 6.35% | 3.95% | 21.23% | 17.87% | 11.19% | 12.54% |
| | $\Delta T$ | 10.16% | 11.75% | 21.44% | 20.95% | 7.01% | 11.58% | 19.95% | 15.41% |
| | $\Delta I$ | 35.89% | 26.88% | 29.48% | 36.28% | 37.86% | 28.57% | 28.48% | 31.42% |
| | $\Delta S$ | 1.76% | 20.29% | 9.02% | 3.99% | -6.94% | 10.16% | 8.99% | 7.09% |
| | $\Delta R$ | -23.09% | -26.24% | -25.13% | -25.52% | -26.12% | -28.71% | -28.88% | -26.22% |
| 2005~2010 | $\Delta TM$ | –0.49% | 17.41% | 7.13% | 3.21% | 13.33% | 7.27% | 8.46% | 7.35% |
| | $\Delta T$ | 26.00% | 14.29% | 23.40% | 24.65% | 18.20% | 24.22% | 22.53% | 22.55% |
| | $\Delta I$ | 38.57% | 23.43% | 28.61% | 21.69% | 30.61% | 24.63% | 28.97% | 28.66% |
| | $\Delta S$ | 0.08% | 13.68% | 10.29% | 20.83% | 0.64% | 10.78% | 10.37% | 9.15% |
| | $\Delta R$ | -17.25% | -31.18% | -28.49% | -27.56% | -29.49% | -31.09% | -29.26% | -27.49% |
| 2010~2015 | $\Delta TM$ | 26.47% | -1.81% | 22.39% | 24.67% | 21.74% | 18.64% | 16.93% | 19.38% |
| | $\Delta T$ | 5.98% | 10.47% | 7.47% | 7.27% | 9.43% | 12.45% | 12.77% | 9.35% |
| | $\Delta I$ | 23.73% | 36.18% | 28.55% | 29.36% | 25.01% | 25.94% | 26.25% | 27.46% |
| | $\Delta S$ | 14.19% | –3.96% | 5.48% | 6.77% | 14.38% | 12.42% | 16.08% | 9.79% |
| | $\Delta R$ | -28.85% | -8.71% | -28.48% | -31.15% | -29.45% | -29.85% | -27.98% | -27.12% |
| 2015~2018 | $\Delta TM$ | 25.19% | 27.07% | 25.66% | 21.80% | 0.42% | 16.34% | 28.53% | 21.87% |
| | $\Delta T$ | 4.86% | 5.34% | 4.73% | 8.61% | 9.44% | 13.65% | 4.32% | 7.01% |
| | $\Delta I$ | 24.02% | 13.12% | 30.57% | 28.73% | 19.70% | 23.41% | 15.06% | 22.94% |
| | $\Delta S$ | 13.49% | 21.26% | 9.39% | 10.02% | 10.73% | 18.48% | 20.61% | 14.67% |
| | $\Delta R$ | -29.64% | -33.20% | -28.61% | -29.39% | -7.86% | -28.11% | -31.48% | -27.67% |
| Overall contribution through 2000~2018 | $\Delta TM$ | 17.53% | 16.99% | 14.66% | 11.96% | 17.92% | 15.58% | 15.98% | 15.76% |
| | $\Delta T$ | 12.01% | 12.19% | 16.15% | 17.44% | 13.85% | 15.75% | 15.40% | 14.82% |
| | $\Delta I$ | 31.15% | 26.95% | 28.89% | 28.00% | 33.53% | 25.03% | 25.00% | 28.18% |
| | $\Delta S$ | 7.33% | 14.84% | 11.69% | 13.60% | 5.56% | 13.22% | 13.97% | 11.50% |
| | $\Delta R$ | -25.30% | -29.03% | -28.61% | -29.01% | -29.14% | -30.43% | -29.64% | -28.67% |

ΔTM declined from 12.54% (2000~2005) to 7.35% (2005~2010), increased to 19.38% (2010~2015), and then reached 21.87% (2015~2018). The contribution of TM to TEG is stable at 14.66%~17.92%, except for Central China (11.96%), with an overall contribution of 15.76%, indicating that TM has a catalytic effect on TEG, and enhancing TM is a crucial way to promote tourism development.

ΔR shows a negative effect on TEG, and the degree of adverse effect increases slowly from 26.22% to 27.67%. The overall contribution rate was 28.67%. Reception is defined as the ratio of the resident population to the number of tourists. This shows that on the premise that the permanent resident population remains basically unchanged, the contribution to TEG can be effectively increased by expanding the scale of tourists.

## Discussion

### Regression results of tourism mobility on tourism economic growth

This study briefly analyzes the regression results of the traditional and spatial panel data model. However, the spatial autocorrelation test results of TEG show an overall trend of fluctuating and increasing spatial correlation, especially with 2009 as the abrupt change point and a significant increase in the degree of agglomeration. Therefore, the article discusses the results of the spatial panel data model in detail, and the primary purpose of analyzing the traditional panel data model is to compare it with the spatial econometric results.

The regression results of the spatial econometric model show that both TM and TA have a significant positive impact on TEG, which verifies the hypothesis we proposed above. This result is also consistent with Wu et al. [93] and Perboli et al. [94]. In contrast, TP and TH have no significant impact on TEG. However, previous studies have also shown that the spatial spillover effect of tourism can significantly affect the TEG [27–29]. Therefore, the impact of TP and TH on TEG remains to be further confirmed.

According to the decomposition results, TM will promote the growth of the local tourism economy but will have a negative impact on neighboring provinces, which indicates a more obvious competition in tourism development among provinces. The increase in mobility in a particular place under a given number of tourists will lead to a diversion of tourists, which will have a negative impact on neighboring regions. Therefore, the tourism industry should also pay attention to the competitive situation in the surrounding areas. The development of tourism focus not only on improving local tourism mobility but also on neighboring areas. Both TP and TH manifest substantial spatial spillover effects. The increase in TP and TH in neighboring areas will produce positive effects, making local areas attach importance to the development of tourism resources and enhancing tourism attraction. TA has a significant positive contribution to TEG, which is consistent with the conclusion of Yang and Wong [27]. However, the spatial spillover effects of TA on TEG are not significant, which may be related to the fact that air traffic does not depend on adjacent spaces.

## Analysis of influencing factors' contribution rate to tourism economic growth

**TM and ΔTM.**   The ΔTM in North, Central, Southwest, and South China all show a trend of "falling and rising." It should be noted that the ΔTM in North China was negative from 2005 to 2010, mainly due to the significant decline in TM in Tianjin and Hebei. The improvement in the transportation infrastructure has a significant impact on TM in Central and Southwest China. The opening of high-speed railroads is a fundamental reason for the fluctuation in ΔTM. For South China, due to the implementation of the overnight visitor count statistics in the tourism statistics system of Guangdong in 2015~2018, the number of tourists decreased significantly compared to 2010~2015, which in turn led to a significant weakening of the ΔTM. In contrast to the regions mentioned above, the ΔTM in Northeast China shows a trend of "rising and falling" changes. From 2010 to 2015, the contribution of TM to TEG in Northeast China declined and was negative. The main reason is the overall decline of the regional economy in the Northeast region at this stage. In 2014 and 2015, the GDP growth rates of Northeast China were 4.23% and -0.84%, respectively, ranking second and last among the seven regions in China during the same period. At the same time, the Northeast region began to carry out statistical "squeeze water" at this stage, which caused obvious fluctuations in the scale of tourists. Therefore, the downturn in the regional economic environment and stricter tourism statistics have negatively affected the contribution of tourism mobility to tourism economic growth. However, since 2016, China has put forward the " all-for-one tourism" policy. Provinces began to pay more attention to the role of tourism in regional economic growth. All-for-one tourism policies and new management systems have led to the continuous improvement of TM in Northeast China from 2015 to 2018, and the contribution to TEG has increased significantly compared with 2010–2015. The ΔTM in East China gradually increased from 6.35% to 25.66%, which is related to the opening of the high-speed railroad network in 2010, leading to a significant increase in TM. Northwest China has made the tourism industry a key point for economic growth, and its tourist reception and transportation construction levels have been rapidly improved under the impetus of the all-for-one tourism strategy.

***Traffic* and ΔT.**   The contribution of ΔT to TEG generally shows a downward trend. However, during the same period, *Traffic* showed a gradual upward trend. In 2018, it increased by 258.72% compared with 2000. Among them, it increased by 35.61% from 2000 to 2005, increased by 91.36% from 2005 to 2010, increased by 24.83% from 2010 to 2015, and increased by 10.73% from 2015 to 2018. From this, it can be judged that there may be a "threshold" in the transportation infrastructure. When the stock of transportation infrastructure in China reaches a certain level, the accumulation of transportation infrastructure cannot improve the contribution to the TEG. The role of transportation infrastructure in influencing tourists' decisions and determining TM cannot be ignored. However, its contribution rate gradually decreases as transportation facilities are gradually improved and regional accessibility differences narrow. The ΔT is 14.82% during the examination period, in which the contribution rate of *Traffic* to TEG in East China (16.15%), Central China (17.44%), Southwest China (15.75%), and Northwest China (15.40%) is higher than that in North, Northeast and South China. This is mainly because Central China and East China are the regions with the largest passenger turnover in China. From 2000 to 2018, the average passenger turnover in Central China and East China was 118.988 billion person-kilometers and 84.595 billion person-kilometers, respectively. The Southwest China and Northwest China are among the regions with the fastest growth in passenger turnover in China, increasing by 3.13 times and 1.77 times respectively, ranking first and second in all regions.

***Industry* and ΔI.**   The tertiary industry consists of transportation, warehousing and postal industry, information transmission, real estate industry, financial industry, wholesale and retail industry, accommodation and catering industry, etc. Tourism is only a part of it. The per capita added value of the tertiary industry reflects the degree of development of the service industry in various regions, and this indicator has achieved a relatively large increase in terms of changing trends. It increased from 3,653 yuan in 2000 to 34,969 yuan, an increase of 8.57 times. The contribution of ΔI to TEG has gradually declined, mainly due to the slowdown in the growth rate of the per capita added value of the tertiary industry. The growth rate dropped from 91.30% in 2000–2005 to 34.35% in 2015–2018. The contribution of ΔI to TEG in North China, South China, Northwest China, and Southwest China is consistent with the national trend. Northeast China, East China, and Central China show different trends. Especially in the Northeast region, the contribution of ΔI to TEG has dropped significantly. The overall contribution rate of *Industry* reached 28.18%, indicating that the quality of tertiary industry development has a vital role in promoting TEG. ΔI is generally stable in East and Central China and declines significantly in Northeast China, which may be related to the deceleration of tertiary industry development, as the data show that the added-value of tertiary industry per capita in Liaoning, Heilongjiang, and Jilin increased by 93.04%, 75.15% and 90.43% from 2010 to 2015, while it only grew by 0.63%, 39.88% and 23.18% from 2015 to 2018. Central China was inconsistent with the overall national trend from 2005 to 2010. This is mainly due to the slow increase in the per capita added value of the tertiary industry during this period, ranking last in all regions. During this period, the industrial structure of Central China was still dominated by industry. In 2010, the average industrial added value accounted for 56.37% of GDP, the highest in all regions of the country. East China was inconsistent with the overall national trend in 2015–2018. The main reason is that the proportion of the tertiary industry in Fujian and Jiangxi in the region has not exceeded 50%, and there is a large room for optimization and improvement of the industrial structure. Therefore, the growth rate of the added value of the tertiary industry per capita exceeds the previous stage, and the contribution of ΔI to TEG is still rising.

***Structure* and ΔS.**   The share of tertiary industry in tourism in Beijing and Tianjin increased significantly from 2010 to 2018 compared to 2000, leading to the rapid growth of ΔS

in North China. The ΔS in Northeast China was -3.96% from 2005 to 2010, mainly since the growth rate of tertiary industry in Heilongjiang and Liaoning lagged behind that of the tourism industry. The ΔS in East, Central, and Southwest China is relatively stable, indicating that tourism and tertiary industry maintain a coordinated development. The ΔS in South China has achieved a shift from negative to positive growth. As the economic volume of Guangdong accounts for a large proportion in South China and the growth rate of tourism significantly lags behind the development rate of the tertiary industry, it leads to a low ΔS in South China from 2000 to 2010. The opening of high-speed rail provides new opportunities for tourism development, and the ΔS in South China gradually increased to 14.38% and 10.73% in 2010~2018. The ΔS in Northwest China has been increasing, which suggests that the tourism economy is the primary driver of tertiary industry growth. The continuous growth of the ΔS contribution to TEG is partially consistent with the findings of Chang et al. [95], De Vita and Kyaw [96], and Zuo and Huang [97]. The higher Structure is, the greater the contribution of ΔS to TEG. However, the literature above mentioned also pointed out that ΔS has a turning point. For example, Zuo and Huang [97] found that this value in China is 8.25%.

***Reception* and ΔR.** The ΔR has a negative impact on TEG. Zuo and Huang [97] used the ratio of tourist arrivals to the permanent resident population to characterize tourism specialization in a study evaluating China's tourism-oriented economic growth. Before reaching the inflection point of 30.34 (that is, the tourism reception effect value is 0.03), this indicator has a significant positive impact on TEG. From 2000 to 2018, the tourism reception effect value dropped from 1.47 to 0.11, still less than 0.03. Therefore, the results of our study also partially confirm the research of Zuo and Huang [97]. While expanding the scale of tourists, various regions should also pay attention to the "inflection point" of the Reception value. When the inflection point is reached, the larger the scale of tourists is, the smaller the contribution to the TEG. However, the ratio of regional population to tourist decreases from 1.47 to 0.11 during the period from 2000 to 2018, indicating that not only the number of tourists should be taken into account, but also the quality of the tourism and the per capita tourism consumption should be attached importance to the TEG. ΔR is relatively stable, among which the southwest and northwest China have the most significant negative contribution to the TEG, indicating that the growth rate of the number of tourists received in the above regions is higher than that of other regions.

## Conclusions

This paper proposes the concept of TM based on the hydrodynamic equation, constructs an econometric model of TEG with TM as the core explanatory variable, explores the direct and indirect effects of TM on TEG, measures the specific contribution of each influencing factor using the LMDI decomposition, and draws the following conclusions.

1. The TM in China has maintained rapid growth for a long time. However, there are differences in the rate of growth in different regions. East China and North China are Leading Area, with the highest average tourism mobility, but the smallest average annual growth rate; Central China, South China, Northeast China, and Southwest China are Stable Area, with the middle average TM and average annual growth rate; Northwest China is Potential Area, which has the smallest average TM, but the largest average annual increase. The TM in each region only showed a significant positive spatial correlation in 2016–2018. The space-time pattern is constantly changing over time. The high-value areas and high-value areas of TM increased significantly, while the low-value areas and low-value areas decreased significantly. The local spatial autocorrelation results of TM are stable, and various agglomeration states are stably distributed in some provinces.

2. The regression results of the traditional panel data model and the spatial panel data model both show that TM has a significant positive effect on TEG. Under the premise of considering the spatial effect, the improvement of TEG in a province by TM will have a negative impact on the adjacent province.

3. Applying the LMDI decomposition method, the TEG is decomposed into *TM*, *Traffic*, *Industry*, *Structure*, and Reception. The results show that the contribution of TM and Structure to TEG showed an upward trend, with average annual contribution rates of 15.76% and 11.50%, respectively. It indicates that improving TM is a crucial way to promote tourism development. The contribution of the *Traffic* and *Industry* to TEG generally showed a downward trend, with average annual contribution rates of 14.82% and 28.18%, respectively. The Reception has a negative impact on the TEG, but it is still a positive contribution, with an average annual contribution rate of 28.67%. The five types of effects of TEG decomposition were different due to regional differences.

The main contributions of this study are as follows: (1) Based on fluid mechanics, we constructed an indicator of TM. We comprehensively consider the impact of tourist arrivals and transportation infrastructure on TEG, which is rarely proposed by scholars in the literature. Our research enriches the research on the influencing factors of TEG. (2) We analyze the influence of TM on TEG based on the econometric model, which highlights the importance of TM. Moreover, we found that TM has negative spatial overflow.(3) Based on the LMDI method, we decompose TEG into five major effects, rather than just considering traditional variables such as human input, capital input, and tourism resource input. Our study further enriches the research on the influencing factors of TEG.

Based on our findings above, we draw the following policy implications. To improve TEG, late-developing regions should improve TM by building large-scale tourism transportation infrastructure, promoting destination marketing to attract tourists, and paying attention to the possible negative effects of increased TM in neighboring regions. At the same time, the improvement of TM should be emphasized at different stages. The threshold effect of tourism transportation infrastructure should also be fully considered. After the transportation infrastructure reaches a certain stock, its contribution to TEG will decrease. At this time, expanding the scale of tourists should become the main tourism development policy.

There are still some limitations in this study. It is difficult to directly collect data on the inflow and outflow of tourist between certain provinces. Therefore, we only select inflow of tourists as the primary data and do not consider the influence of the tourists' outflow on TM. In fact, increased transport accessibility will not only expand the inflow of tourists but also affect the outflow of tourists. Therefore, the superposition effect of traffic and tourist inflow/outflow should be considered comprehensively to improve the scientific rationality of TM measurement. This study lacks comparative studies across multiple countries. The research in our study may show differentiated findings for developed or less developed countries. When constructing the econometric model, we mainly consider TM as the core explanatory variable, and only select human input and capital input, and air traffic related to traffic as control variables from the perspective of the economic growth model. In the future, the theory and practice of TM will be further explored with multivariate data to form a more rigorous and systematic cognitive framework.

## Supporting information

**S1 Fig. Map of the seven regions.**
(DOCX)

**S1 File. Research data.**
(XLSX)

## Author Contributions

**Conceptualization:** Jun Liu, Yun Tong.

**Data curation:** Fan Yu.

**Formal analysis:** Fan Yu.

**Funding acquisition:** Jun Liu.

**Investigation:** Yun Tong.

**Methodology:** Jun Liu, Yun Tong.

**Resources:** Jun Liu.

**Software:** Mengting Yue.

**Writing – original draft:** Mengting Yue.

**Writing – review & editing:** Fan Yu.

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
