## [Decision Letter · Decision Letter 0]

23 May 2022

PONE-D-22-06360The Contribution of Tourism Mobility to Tourism Economic Growth in ChinaPLOS ONE

Dear Dr. Yu,

Thank you for submitting your manuscript to PLOS ONE. After careful consideration, we feel that it has merit but does not fully meet PLOS ONE’s publication criteria as it currently stands. Therefore, we invite you to submit a revised version of the manuscript that addresses the points raised during the review process.

Both reviewers recommended major revision. I agree to their comments. Please upgrade your manuscript by addressing the comments from the reviewers.

We look forward to receiving your revised manuscript.

Kind regards,

Hironori Kato, Dr. Eng.

Academic Editor

PLOS ONE

Journal Requirements:

Reviewers' comments:

Reviewer's Responses to Questions

**Comments to the Author**

1. Is the manuscript technically sound, and do the data support the conclusions?

Reviewer #1: Yes

Reviewer #2: Partly

2. Has the statistical analysis been performed appropriately and rigorously? 

Reviewer #1: Yes

Reviewer #2: Yes

3. Have the authors made all data underlying the findings in their manuscript fully available?

Reviewer #1: Yes

Reviewer #2: No

4. Is the manuscript presented in an intelligible fashion and written in standard English?

Reviewer #1: Yes

Reviewer #2: Yes

5. Review Comments to the Author

Reviewer #1: Thank you for the opportunity to review the manuscript. This paper is aiming to examine the contribution of variables to TEG, focusing on tourism mobility. The authors also used a statistical method to verify the model or variables and provided validity. Despite those efforts being seemingly great, the authors can further improve the manuscript with revisions. The reviewer would like to comment as follows and hope these comments will be helpful for the authors.

1. The purpose and the implications of this study

The authors mentioned the purpose of this study as follows. “This paper put forward the concept of tourism mobility based on the theory of fluid mechanics, explores its impact on TEG, and analyzes the contribution of each influencing factor to TEG which will provide implications for the high-quality development of tourism economy.” Please clearly define “high-quality development.” What is the definition of quality? What are the relationships between quality and tourism mobility? As for the implications mentioned in the Conclusion, it is unclear which is of high quality. The authors only mentioned that improving tourism mobility is crucial for promoting tourism development. The findings of this study were comprehensible; however, the authors should clearly define high quality because that is one of the aims of this study.

2. Econometric model and LMDI decomposition

In section 1.2.3, the authors defined Equation (3) as an econometric model and mentioned each variable in the first paragraph. However, some variables are inconsistent or cannot be found in Equation (3). For example, tourism mobility (TM) cannot be found in the equation. Mobility in the equation seems to mean TM; however, their meanings are different. TP, TH, and TA in the paragraph cannot be found in the equation. TK in the equation was not mentioned in the first paragraph. Please check the equations carefully and revise them.

In 1.2.4, there are the same issues as above. For example, traffic effects (Traffic) cannot be found in Equations (5)-(8). Is it the same as Transportation? Industry cannot be found in the equation. Additionally, it was mentioned that tourism mobility only considers land transportation in 1.2.3. In 1.2.4, Transportation indicates the weighted road length. What is the reason behind employing only road length as the weight value?

3. Sections 2.1 and 4.1: Regions

This paper segmented China into seven regions and introduced each region with the data in section 2.1. The authors also introduce each transportation network region, mobility, and policies regarding transportation in 4.1. However, the introduction of each region in 4.1 should be provided in 2.1. Those situations are the presumptions of the analysis, not resulting from the estimation.

The explanation for Figure 2 should be reconfirmed and revised appropriately. The authors mentioned that tourism mobility in South China grew rapidly; however, the classification defined in Group (2) includes South China, Southwest China, Central China, and Northeast China. If South China means Group (2) in the sentence, it should describe these regions instead of “South China” only.

4. Sections 3.4 and 4.3: Factors

In section 3.4, the influencing factors in each period are considered with the estimation result. However, some factors were discussed by comparing the result between 2000~2005 and 2015~2008; others were only with the overall contribution rate, meaning 2000~2018. What is the reason for considering the factors differently? It seems the authors want to mention (inversed) V shape change. If so, the authors should mention the meaning of (inverse) V change in 3.4 and discuss it in 4.3 using the term “(inverse) V change.”

As for the factor of host-guest interaction, the result showed a negative effect on TEG. What does the effect of host-guest interaction mean to TEG? Please discuss this point in 4.2.

Table 8 should be revised for the comprehension of readers. At the bottom of the table, the overall estimation result is described as 2000~2018; however, this description is confusing. For example, it is more understandable if named “Overall contribution through 2000~2018.”

In 4.3, the authors mentioned that the support for the tourism industry has weakened in Northeast China; however, there was no evidence for it. This is an insufficient explanation and a jump of logic. Please provide evidence to support the logic.

5. Recommendation: words and definitions

For the sake of clarity and appropriateness, the authors should consider changing some words and definitions. In Figure 1, tourist sources seem better changed to tourist origins. In 1.2.4, the authors defined host-guest interactions effects; however, host-guest interactions are defined and developed on the scale in other areas, such as residents’ attitudes toward tourism development. The host-guest interactions in this study seem to mean the ratio of tourists per resident, and it is more appropriate to change the variable’s name. Additionally, please consider clarifying what the host-guest interaction effect means to TEG?

6. Recommendation: Shifting the future assignments from the Discussion section to Conclusion.

The authors mentioned the limitation and the future assignments of this study at the end of the Discussion section. It would be better to mention those points at the end of the Conclusion. The reviewer would like to recommend shifting the last paragraph (future assignment) to the end of the Conclusion.

7. Minor comments: Spelling errors and inconsistent terms

The authors should check and revise the misspelled words and inconsistencies throughout the manuscript. For example, GDP (ervice) in Equation 8 should be corrected to GDP (service).

Reviewer #2: Review of “The Contribution of Tourism Mobility to Tourism Economic Growth in China”

The manuscript has the potential to make various contributions to the tourism-related policy literature but should be strengthened by addressing several critical deficiencies (outlined below).

(a) Introduction

o It would be helpful to briefly explain the connections between tourism mobility (TM) and tourism economic growth (TEG) from the geospatial perspective early in the paper and return to this when discussing the policy implications of the research findings.

o Although the authors have given some explanation of the current situation of economic development in China, it is currently insufficient. More context is needed regarding the present situation of the tourism industry, TM, and TEG.

o The research question(s) and main hypothesis or hypotheses should be included in this section.

o The second and third paragraphs of this section should be restructured as a new, separate literature review section. The authors are encouraged to include more previous studies and discuss them in greater depth in the literature review, as the literature reviewed in the original version of the manuscript is insufficient.

(b) Econometric model

o A few words or a sentence should be sufficient to describe the spatial weight matrix used in equation 4.

(c) Data sources

o The last paragraph of this section (about the seven regions of the study area) should be moved to the Appendix. Including a map inset of these areas would be helpful here, as well.

(d) Spatial evolution

o It would be helpful to include a LISA cluster map of to help readers better understand the spatial patterns described here.

(e) Discussion

o This section does not sufficiently analyze the authors’ findings in the context of previous studies. How and why does this study differ from previous research? Relatedly, there are some similarities between this study’s results and those of previous research—what is the unique contribution of this paper? Please emphasize these differences more clearly.

o It seems that the policy implications of this manuscript are limited. Thus, the authors are encouraged to derive policy implications or recommendations based on their findings. Such implications need not be restricted to Chinese tourism-related policies but may also be applicable to other countries with conditions similar to China. The authors are encouraged to clarify these issues in this section.

o Theoretical and practical contributions should be highlighted more explicitly in this section.

o The authors are encouraged to expand their discussion of the research limitation(s), as this issue is currently insufficiently explained.

Final comments

Given the several critical deficiencies pointed out above, this paper is not ready for publication in its current form. I recommend a major revision, with resubmission possible if the authors are able to logically and satisfactorily address these deficiencies.

6. PLOS authors have the option to publish the peer review history of their article (what does this mean?). If published, this will include your full peer review and any attached files.

Reviewer #1: No

Reviewer #2: No

---

## [Author Response · Author response to Decision Letter 0]

7 Aug 2022

Letter to Reviewer ’s Comments

Dear anonymous reviewer,

Thanks deeply for your comments. We had studied all of your constructive and helpful comments very carefully and tried to incorporate all of them into the current version of the paper, it definitely improved the paper a great deal. 

Please find the point-by-point response to each of your comment and more detailed descriptions of how we did that below. For ease of distinction, your originals comments are underlined, and our responses to those comments and amends are shown in regular font. The sentences from manuscript are in italic font.

*For your convenience, the page number and section number listed below are based on the “Revised Manuscript with Track Changes”.

Reviewer #1: 

Thank you for the opportunity to review the manuscript. This paper is aiming to examine the contribution of variables to TEG, focusing on tourism mobility. The authors also used a statistical method to verify the model or variables and provided validity. Despite those efforts being seemingly great, the authors can further improve the manuscript with revisions. The reviewer would like to comment as follows and hope these comments will be helpful for the authors..

1.The purpose and the implications of this study

The authors mentioned the purpose of this study as follows.“This paper put forward the concept of tourism mobility based on the theory of fluid mechanics, explores its impact on TEG, and analyzes the contribution of each influencing factor to TEG which will provide implications for the high-quality development of tourism economy.” Please clearly define “high-quality development.”What is the definition of quality? What are the relationships between quality and tourism mobility? As for the implications mentioned in the Conclusion, it is unclear which is of high quality. The authors only mentioned that improving tourism mobility is crucial for promoting tourism development. The findings of this study were comprehensible; however, the authors should clearly define high quality because that is one of the aims of this study. 

Response 1：Thank you very much for your constructive comment. High-quality development is proposed by the Chinese government in 2017. It shows that China's economy shifts from a high-speed growth stage to a high-quality development stage. Since 2020, researches on the high-quality development of tourism have begun to appear (Liu and Han, 2020;Shi et al.,2021;Lu, 2022). However, there is still no unified definition of high-quality tourism. Some studies use specific indicators such as the average consumption level of tourists or the conversion efficiency of tourism resources to measure the level of high-quality tourism. Some other studies also build an index system to measure the level of high quality. No matter what the widely accepted definition of high-quality tourism development is, it implies that the development of tourism should shift from the pursuit of scale to the requirement of quality. Whether tourism mobility can be used as an indicator to measure the high-quality development of tourism in this study is still inconclusive. However, tourism mobility is a comprehensive indicator that comprehensively considers the flow of tourists and regional transportation infrastructure. At the same time, in the process of LMDI decomposition, tourism economic growth is decomposed into different effects. These are not only the total output, but also the quality of tourism. To sum up, the research conclusions put forward in the introduction part of this study have reference value for the high-quality development of regional tourism. Since the focus of this study is not on the high-quality development of the tourism economy, we believe that specifically defining the high-quality development of tourism may make the research go off-topic. Therefore, to avoid ambiguity, we decided to delete the expression “high-quality development”. Thanks still for your suggestion!

2. Econometric model and LMDI decomposition

In section 1.2.3, the authors defined Equation (3) as an econometric model and mentioned each variable in the first paragraph. However, some variables are inconsistent or cannot be found in Equation (3). For example, tourism mobility (TM) cannot be found in the equation. Mobility in the equation seems to mean TM; however, their meanings are different. TP, TH, and TA in the paragraph cannot be found in the equation. TK in the equation was not mentioned in the first paragraph. Please check the equations carefully and revise them.In 1.2.4, there are the same issues as above. For example, traffic effects (Traffic) cannot be found in Equations (5)-(8). Is it the same as Transportation? Industry cannot be found in the equation. Additionally, it was mentioned that tourism mobility only considers land transportation in 1.2.3. In 1.2.4, Transportation indicates the weighted road length. What is the reason behind employing only road length as the weight value? 

Response 2: Thank you very much for your careful reading and reminder. First, we are sorry for the confusing equation and inconsistent variable names. We have corrected this inconsistent variable names and checked for them all throughout the paper again. The one-to-one correspondence between variable names and variable meanings is as follows：

TEG——tourism economic growth

TM——tourism mobility

TH——human capital in the tourism industry

TA——passenger traffic by the airport

TP——physical capital in the tourism industry

Traffic——the cumulative traffic effects by LMDI decomposition

Industry——the effects of the tertiary industry by LMDI decomposition

Structure——the structural effects of the tourism industry by LMDI decomposition

Reception——the reception effects by LMDI decomposition

——the contribution of TM to TEG 

——the contribution of the Traffic to TEG 

——the contribution of the Industry to TEG

——the contribution of the Structure to TEG

——the contribution of the Reception to TEG

Second, the reason for employing only land transportation to measure the tourism mobility is that China's railway and road passenger traffic accounts for the vast majority of the total passenger traffic. The average share of rail and road passenger traffic from 2000 to 2018 was 97.6%. We added a few sentences “China's railway and road passenger traffic accounts for the vast majority of the total passenger traffic. Furthermore, we were unable to calculate weighted air and water transportation infrastructure lengths, so we only consider the land transportation infrastructure data.” in section 2.2.1 (page 7) to explain the reason for employing only road length as the weight value.

3. Sections 2.1 and 4.1: Regions

This paper segmented China into seven regions and introduced each region with the data in section 2.1. The authors also introduce each transportation network region, mobility, and policies regarding transportation in 4.1. However, the introduction of each region in 4.1 should be provided in 2.1. Those situations are the presumptions of the analysis, not resulting from the estimation.

The explanation for Figure 2 should be reconfirmed and revised appropriately. The authors mentioned that tourism mobility in South China grew rapidly; however, the classification defined in Group (2) includes South China, Southwest China, Central China, and Northeast China. If South China means Group (2) in the sentence, it should describe these regions instead of“South China”only. 

Response 3: Thank you very much for your constructive comment. We rewrite the section 3.1 (page 15) Spatiotemporal evolution characteristics of tourism mobility. First, we moved the content as you suggested. Second, we revise the content of section to explain the Figure 2. We divided into three paragraphs to analyze the temporal evolution of tourism mobility in the three types regions. For your convenience, we present this section as below:

Limited by space, Table 1 only shows the results of TM over five years. During the study period, TM increased from 56~12745 p visitors /km to 382~18865 p visitors /km, with an average annual growth rate between 2.20% and 13.46%. According to the average value of TM (Figure 2), the study areas are divided into the following three types.

 (1) “Leading Area”，including East China and North China, ranked first and second in all regions.Their TM increased from 2679.39 and 1884.34 p visitors/km in 2000 to 5859.93 and 5209.94 p visitors/km in 2018. However,their annual average growth rates were 5.07% and 6.43%, respectively, ranking first and second from the bottom in all regions. East China is located on the coast, relying on superior natural conditions and an economic foundation, and its regional transportation system is relatively complete.Therefore, it has formed a number of advantageous tourist resource gathering areas, and has become the main tourist destination of inbound tourists in China, and its mobility has long ranked first in the country. As a political and economic center, Beijing has become a tourist attraction for domestic and inbound tourism with a large number of historical and cultural tourism resources. It also drives the joint development of the tourism industry in North China with the Beijing-Tianjin-Hebei urban agglomeration as the core, making North China the second largest core area of TM after East China.

(2)“Stable Area”,including South China, Southwest China, Central China, and Northeast China, ranked third to sixth in all regions.Their TM increased from 903.57p visitors/km, 695.15p visitors/km, 632.06p visitors/km, 493.33 p visitors/km in 2000 to 2626.11p visitors/km, 2754.97p visitors/km, 2857.88p visitors/km, 2244.68 p visitors/km in 2018.The average annual growth rates were 6.58%, 8.81%, 9.06% and 9.38%, respectively.TM in South China grew rapidly during 2005~2015, while it has gradually slowed down in recent years. This is mainly due to the construction of the early transportation system in South China, which increased tourist mobility. After the basic construction of facilities, the incremental tourist inflows decreased, and the overall growth remained stable. Central China has become one of the core transportation hubs under its location and has driven regional tourism development, becoming a central province in the second echelon of TM. Due to geographical restrictions, Northeast and Southwest China are less connected to the transportation network than coastal areas, resulting in relatively low levels of TM. Northeast China focuses on the development of heavy industry but pays little attention to tertiary industry, and tourism infrastructure construction and resource development are relatively weak, which leads to low TM. There are many mountains in Southwest China, and its early traffic development level lags behind.With the opening of the Chengdu-Chongqing high-speed railway, and Chengdu-Guizhou high-speed railway, and the development of the air transportation industry, the land and air transportation layout in Southwest China is becoming increasingly mature. Southwest China actively developed its resources, and the tourist inflow increased from 145 million (2000) to 2.994 billion (2018), with an average value of TM catching up with that of southern China during 2016~2018. 

(3)“Potential Area”, including Northwest China, ranks last in terms of average tourist mobility. Its TM increased from 282.01 p visitors/km in 2000 to 1427.58 p visitors/km in 2018, but its average annual growth rate was 10.01% , ranking first among all regions.As a less developed regions, northwestern China has a poor foundation in economic development and openness to the outside world, and TM has long been at the bottom of the list. Although TM in Northwest China has long been at the bottom of the list, its mobility growth rate leads other regions as tourism infrastructure construction and resource development levels have improved under the active promotion of Western Development policies, the Five-Year Plan, and the Territorial Tourism Strategy. 

Third, in order to more intuitively observe the temporal and spatial change characteristics of TM during the study period, we apply the method of natural breaks to classify the 31provinces and apply the standard deviation ellipse to identify the direction of TM in each province. It read as

To more intuitively observe the temporal and spatial change characteristics of TM during the study period, we apply the method of natural breaks to classify the 31 provinces.Natural breaks classes are based on natural groupings inherent in the data. Class breaks are identified that best group similar values and that maximize the differences between classes. The features are divided into classes whose boundaries are set where there are relatively big differences in the data values. The natural breaks classification method is a data classification method designed to determine the best arrangement of values into different classes. This is done by seeking to minimize each class’s average deviation from the class mean, while maximizing each class’s deviation from the means of the other groups（Chen et al.,2013). We divided the 31 provinces into five categories,highest-value area, higher-value area,medium-value area,lower-value area, and lowest-value area, according to the TM in 2000, 2005, 2010, 2015, and 2018.As shown in Figure 3, (1) Shanghai and Beijing have long been in the highest-value area and higher-value area of TM. Tibet, Qinghai, Ningxia, Xinjiang, Inner Mongolia, Gansu, Jilin, Heilongjiang, Hubei, and Hainan have long been in the lowest-value and lower-value areas.(2) Over time, the number of provinces in the highest-value area and the higher-value area increased significantly, from 2 provinces in 2000 to 12 provinces in 2018. The number of provinces in the lowest-value area and lower-value area significantly decreased, from 26 provinces in 2000 to 12 provinces in 2018; the number of provinces in the medium-value area fluctuated randomly, with the fewest 3 in 2000 and the most 13 in 2015.(3) Except for Shanxi, Northwest China has been in the lowest-value area and the lower-value area for a long time; The TM values in Southwest China have changed greatly. Chongqing and Guizhou have jumped from the lower-value area to the higher-value area,and Yunnan has jumped from the low-value area to the medium-value area.Tibet is relatively stable and has been in the lowest-value area for a long time;South China is relatively stable, but the average value TM in Guangxi has changed greatly, jumping from the lower-value area to the higher-value area;The average TM in Central China has been in the low-value area for a long time.Central China is also relatively stable, and its average of TM has long been located in the lower-value area and the medium-value area. With the exception of Shanghai, which has always been in the highest-value area, the initial value of TM in other provinces in East China has jumped upward. In the Northeast, Liaoning's TM has always been in a leading position, and it has gradually transitioned from a lower-value area to a higher-value area.However,Jilin and Heilongjiang have always been in the lowest-value area and the lower-value area, respectively. Changes in TM in North China are diverse. Beijing has long been located in the highest-value area and higher value area. Inner Mongolia has been in the lowest-value area for a long time. Hebei is in the lower-value area most of the time.Tianjin and Shanxi changed greatly, and finally jumped to the highest-value area and the higher-value area, respectively.

We use the standard deviation ellipse to identify the direction of TM in each province. As shown in Figure 3, the lengths of the minor semiaxis and major semiaxis of the ellipse increased significantly.The growth of the short semiaxis reveals that the degree of dispersion of TM in China's provinces is gradually increasing. This result is basically consistent with the previous analysis conclusions that TM in some provinces shows a more obvious transition trend, which makes the overall dispersion of TM increase.

Fig.3 The spatiotemporal pattern and direction distribution of provincial TM

4. Sections 3.4 and 4.3: Factors

In section 3.4, the influencing factors in each period are considered with the estimation result. However, some factors were discussed by comparing the result between 2000~2005 and 2015~2008; others were only with the overall contribution rate, meaning 2000~2018. What is the reason for considering the factors differently? It seems the authors want to mention (inversed) V shape change. If so, the authors should mention the meaning of (inverse) V change in 3.4 and discuss it in 4.3 using the term “(inverse) V change.”

As for the factor of host-guest interaction, the result showed a negative effect on TEG. What does the effect of host-guest interaction mean to TEG? Please discuss this point in 4.2.

Table 8 should be revised for the comprehension of readers. At the bottom of the table, the overall estimation result is described as 2000~2018; however, this description is confusing. For example, it is more understandable if named“Overall contribution through 2000~2018.”

In 4.3, the authors mentioned that the support for the tourism industry has weakened in Northeast China; however, there was no evidence for it. This is an insufficient explanation and a jump of logic. Please provide evidence to support the logic. 

Response 4: Thank you very much for your insightful comment that helps us to improve this manuscript. 

First, there are slight differences in how we describe the estimate result of each influencing factor. This is because some factors have a V-shaped trend of rising first and then falling, or first falling and then rising. However, some other factors have been maintaining the same trend of change, rising or falling. Therefore, we describe the former in more detail. Following your suggestion, we have marked the period next to the description of each value for ease of reading. 

Second, we think it is inaccurate to use the expression V-shaped or inverted V-shaped to represent the trend of rising first and then falling.We have revised the relevant content and provided a more detailed explanation of the trend of each variable in section 4.3 and section 5.2. 

Third, we change the define “host-guest interaction effect” to “Reception Effects” according your suggestion 5. Our results showed a negative effect of reception effects on TEG. Since the resident population changes less in a period of time, it means that when the value of the Reception is larger, the actual number of tourists is less.Therefore, the Reception is an inverse indicator, that is, it has a negative impact on TEG.We give more explanation about the results in section 5.2, which reads as

Reception and : The has a negative impact on TEG.Zuo and Huang (2017) used the ratio of tourist arrivals to the permanent resident population to characterize tourism specialization in a study evaluating China's tourism-oriented economic growth. Before reaching the inflection point of 30.34 (that is, the tourism reception effect value is 0.03), this indicator has a significant positive impact on TEG. From 2000 to 2018, the tourism reception effect value dropped from 1.47 to 0.11, still less than 0.03. Therefore, the results of our study also partially confirm the research of Zuo and Huang (2017). While expanding the scale of tourists, various regions should also pay attention to the "inflection point" of the Reception value. When the inflection point is reached, the larger the scale of tourists is, the smaller the contribution to the TEG. However, the ratio of regional population to tourist decreases from 1.47 to 0.11 during the period from 2000 to 2018, indicating that not only the number of tourists should be taken into account, but also the quality of the tourism and the per capita tourism consumption should be attached importance to the TEG. The is relatively stable, among which the southwest and northwest China have the most significant negative contribution to the TEG, indicating that the growth rate of the number of tourists received in the above regions is higher than that of other regions. 

Fourth, we renamed the description in the bottom of the table 8 to “Overall contribution through 2000~2018”. 

Fifth, we explain the reason for “the in Northeast China shows a trend of "rising and falling" changes.” in more details. It reads as

 In contrast to the regions mentioned above, the in Northeast China shows a trend of "rising and falling" changes.From 2010 to 2015, the contribution of TM to TEG in Northeast China declined and was negative. The main reason is the overall decline of the regional economy in the Northeast region at this stage. In 2014 and 2015, the GDP growth rates of Northeast China were 4.23% and -0.84%, respectively, ranking second and last among the seven regions in China during the same period. .At the same time, the Northeast region began to carry out statistical "squeeze water" at this stage, which caused obvious fluctuations in the scale of tourists. Therefore, the downturn in the regional economic environment and stricter tourism statistics have negatively affected the contribution of tourism mobility to tourism economic growth. However, since 2016, China has put forward the " all-for-one tourism" policy. Provinces began to pay more attention to the role of tourism in regional economic growth. All-for-one tourism policies and new management systems have led to the continuous improvement of TM in Northeast China from 2015 to 2018, and the contribution to TEG has increased significantly compared with 2010-2015. 

5. Recommendation: words and definitions

For the sake of clarity and appropriateness, the authors should consider changing some words and definitions. In Figure 1, tourist sources seem better changed to tourist origins. In 1.2.4, the authors defined host-guest interactions effects; however, host-guest interactions are defined and developed on the scale in other areas, such as residents’ attitudes toward tourism development. The host-guest interactions in this study seem to mean the ratio of tourists per resident, and it is more appropriate to change the variable’s name. Additionally, please consider clarifying what the host-guest interaction effect means to TEG?

Response 5: Thank you very much for your detailed comment. First, we revised the “tourist sources” to “tourist origins” in Figure1.Second, we change the variable name “host-guest interactions effects” to “the reception effects”, which is abbreviated as Reception. Third, we give more explanation for the negative effects of Reception on TEG.We have shown the content in the response 4. 

6. Recommendation: Shifting the future assignments from the Discussion section to Conclusion.

The authors mentioned the limitation and the future assignments of this study at the end of the Discussion section. It would be better to mention those points at the end of the Conclusion. The reviewer would like to recommend shifting the last paragraph (future assignment) to the end of the Conclusion. 

Response 6: Thank you very much for your detailed comment. We moved the paragraph you mentioned to the end of the Conclusion. 

7. Minor comments: Spelling errors and inconsistent terms

The authors should check and revise the misspelled words and inconsistencies throughout the manuscript. For example, GDP (ervice) in Equation 8 should be corrected to GDP (service). 

Response 7: Thank you very much for your careful reading and reminder. We are sorry for the typo. We have corrected this error and checked for typos throughout the paper again.

Reviewer #2: 

The manuscript has the potential to make various contributions to the tourism-related policy literature but should be strengthened by addressing several critical deficiencies (outlined below).

(a) Introduction

oIt would be helpful to briefly explain the connections between tourism mobility (TM) and tourism economic growth (TEG) from the geospatial perspective early in the paper and return to this when discussing the policy implications of the research findings. 

oAlthough the authors have given some explanation of the current situation of economic development in China, it is currently insufficient. More context is needed regarding the present situation of the tourism industry, TM, and TEG.

oThe research question(s) and main hypothesis or hypotheses should be included in this section.

oThe second and third paragraphs of this section should be restructured as a new, separate literature review section. The authors are encouraged to include more previous studies and discuss them in greater depth in the literature review, as the literature reviewed in the original version of the manuscript is insufficient. 

Response a: Thank you very much for your insightful comment that helps us to improve this manuscript. We add the section “Literature Review” and improve the manuscript in introduction and literature review, respectively:

(1)We rewrite the introduction to address those question you mentioned above.We explain the connections between tourism mobility and tourism economic growth in the second and fourth paragraphs. We also add the explanation of the current situation of tourism in China in the third paragraph. In the 4-7 paragraphs, we give the research questions, research methods, and main hypothesis and research contributions. The introduction reads as:

Introduction

In recent years, the tourism industry has maintained rapid development. By 2019, the total number of global tourist trips exceeded 12.3 billion, an increase of 4.6% over the previous year. The total global tourism revenue was US$5.8 trillion, equivalent to 6.7% of global GDP (World Tourism Economy Trends Report (2020)).Tourism has made important contributions to economic growth by increasing employment, improving infrastructure, and accumulating foreign exchange earnings for destinations(Li, et al, 2018).Due to the impact of COVID-19, People's travel is restricted. The total number of international tourists in 2021 decreased by 72% compared with 2019, and international tourism consumption dropped by nearly half compared with 2019 (UNWTO, 2022). 

The above facts remind us that mobility has become an essential feature of tourism activities(Szivas et al., 2003; Liu and Wall, 2006). Tourist from origins to desinations resulting in a series of mobility of information, material, and capital. These mobilities have a great influence on tourist destinations (Urry, 2003; Cárdenas - García et al, 2013;Hannam et al., 2014; Kim et al., 2021).If tourism mobility(TM) stagnates, tourist attractions, reception facilities and transportation facilities built for tourists will be idle. Tourism workers will lose their jobs and tourism economic growth(TEG) will also stagnate.Therefore, studying the impact of TM is necessary and important. 

As one of the important tourist destinations in the world, China's domestic tourism and inbound tourism are developing rapidly. In 2019, the total contribution of China's tourism industry to GDP reached 10.94 trillion yuan, accounting for 11.05% of the total GDP, exceeding the proportion of international tourism in global GDP. A total of 28.25 million people were directly employed in tourism, and 51.62 million people were indirectly employed in tourism. The total employment in tourism accounts for 10.31% of the total employment population in the country (Ministry of Culture and Tourism of China, 2020). However, due to the impact of the COVID-19, the development level of China's tourism industry has not recovered to the level of 2019. In 2021, the total number of domestic tourists in China was 3.246 billion, which is only 54% of that in 2019, and directly leads to a total tourism revenue of 2.92 trillion yuan, which is only 51% of that in 2019.This shows that TM is more important to China's tourism industry.Therefore, we decide to focus on the TM in this study and take China as the research sample.

The top priority of this study is to obtain the right measurement of TM.Transportation infrastructure is an important carrier for the exchange of factors in tourism. Existing studies have confirmed that transportation is a key factor in promoting TEG (Prideaux, 2005; Wang et al., 2012; Massidda and Etzo, 2012).The establishment of the transportation system has an obvious effect on improving the accessibility of tourist destinations and promoting the inflow of the tourist population (Li et al., 2019). However, most existing studies only take tourist arrivals to characterize TM (Pearce, 1979; Gunn, 1988 ; Khadaroo and Seetanah, 2008; Chi, 2014; Liu et al., 2016; Zhang et al.,2019;Saayman and de klerk, 2019). They ignore that the transportation infrastructure is also an important factor affecting the TEG.Therefore, this study redefines TM, which considers both transport infrastructure and tourist arrivals. 

Another important purpose of this study is to explore the effect of TM on TEG. Existing literature analyzes the links between TM and international trade (Keum, 2010; Morley, 2014) or focuses on the relationship between economic growth (Kadir and Karim, 2012; Nuno and Muhammad, 2016). However, less literature has focused on the relationship between TM and TEG.There are two possible reasons for the lack of attention. First, the positive and significant impact of the tourist arrivals and TEG no longer needs to be verified. It is common sense that the more tourists the destination receives, the higher the tourism income. Second, tourist arrivals, as a single indicator to measure TM, is obviously able to affect the TEG. Our measurement of the TM concludes both transport infrastructure and tourist arrivals in this study. Therefore, we decide to explore the contribution of TM to the TEG based on the new measurement for TM.

We first use econometric methods to test whether there is a significant impact of TM on TEG. Considering the positive impact of transport infrastructure on China's TEG (Zhang et al, 2020), we hypothesize that TM has a positive impact on TEG.Previous studies have also shown that the spatial spillover effect of tourism may significantly affect the TEG (Yang et al.,2012; Yang and Fik,2014;Yuming, 2014). Therefore, we further apply the spatial Durbin model to test the impact of TM on TEG.

Moreover, we also use the LMDI (Logarithmic Mean Divisia Index) method to further analyze the contribution of TM to TEG in more detail. The LMDI method is often used to study environmental issues such as energy consumption and carbon emissions (Colinet et al, 2015; Chong et al, 2017). In the field of tourism research, the LMDI method is mostly used to decompose tourism carbon emissions or energy consumption (Pablo-Romero et al, 2021; Zha et al, 2021). There are few studies using the LMDI to analyze TEG. Therefore, we further use the LMDI method to decompose TEG into five influencing factors including the tourism mobility effects (TM), the cumulative traffic effects (Traffic), the effects of the tertiary industry (Industry), the structural effects of the tourism industry (Structure) and the reception effects (Reception), and examine the contribution of TM to TEG.

Different from previous studies, this study makes two contributions to the literature. First,we introduce the related concepts of fluid mechanics to construct the indicator TM. We also consider the superposition effect of tourist arrivals and transportation infrastructure. This deepens the understanding of TM and promotes the integration of interdisciplinary knowledge.Second, we are the first to examine the impact of TM on TEG using econometric models and the LMDI method. This deepens the understanding of the mechanisms that influence TEG. The results of this study also provide a reference for tourism-related policy makers. Regions wishing to develop tourism can achieve TEG by expanding the size of the source market and promoting the construction of transportation infrastructure.

The rest of this study is organized as follows. Section 1 summarizes the relevant literature. Section 2 presents the theoretical framework, methods, and data. Section 3 introduces the spatiotemporal pattern and evolutionary trend of TM. Section 4 analyzes the contribution of TM to TEG from two different perspectives. Section 5 discusses and analyzes the research results. The last section concludes this study.

(2)We add a new section “Literature review” to include more previous studies and discuss them in greater depth. The literature review reads as:

1.Literature review

As the core of tourism activities, TM refers to the mobility of tourists from the origin to the destination, and the stay of tourists in the region (Oppermann, 1995).It is often associated with tourism demand and is measured by tourist arrivals (Song and Witt, 2006).Since the 1970s, many studies have paid attention to the influencing factors and the spatial structure of TM (Pearce, 1979; Gunn, 1988). The existence of regional heterogeneity makes TM affected by many factors, such as infrastructure, income, GDP and cultural distance (Khadaroo and Seetanah, 2008; Chi, 2014; Zhang, 2019). Moreover, it also makes the spatial structure of TM different.Therefore, TM prediction has become one of the research hotspots (Song, 2008). A large body of research has focused on TM forecasting (Saayman and de klerk, 2019), including using a combination and integration of forecasts, using nonlinear methods for forecasting, and extending existing methods to better model the changing nature of tourism data (Saayman and Botha, 2017).The gravity model is an earlier method used to analyze international TM (Durden and Silberman, 1975). Due to its effectiveness in explaining TM (Keum, 2010), gravity models are often used to analyze international tourism service trade. Although the use of gravity models to predict bilateral TM still lacks a corresponding theoretical explanation mechanism, empirical evidence supports the applicability and robustness of gravity models for TM (Morley, 2014). Existing research focuses on examining the movement patterns and spatial structure of international TM in destinations (Lozano and Gutiérrez, 2018), such as the transfer of inbound TM within regions and the influencing factors of inbound TM within destinations (Hwang et al., 2006). There are still few studies on the overall spatial characteristics of TM within destination countries, and the only literature is mainly based on digital footprints or questionnaire data to analyze the spatial structure of TM (Wang et al., 2016; Liu et al., 2021).

Unlike the tourist arrivals indicator, which focuses more on the group flow of people, TM examines a wider range of content, including the flow of people, the flow of materials, the flow of ideas (more intangible thoughts and fantasies), and the flow of technology. (Hannam et al, 2014). Early tourist movement focused more on tourist travel decisions and the resulting movement patterns. Lue et al. (1993) summarized five travel patterns of tourists between destinations. Li et al. (2008) revealed the spatial patterns of TM and tourism propensity in the Asia-Pacific region over the past 10 years. Mcckercher and Lau (2008) took Hong Kong as an example and identified 78 movement patterns and 11 movement styles of TM within the destination. In recent years, with the help of technologies such as GPS, GIS, and RFID, the movement of tourists within scenic spots has attracted attention (Zheng et al, 2017). Research on visitor movement in national parks, theme parks, protected areas, etc. continues to increase (Connell and Page, 2008; Hallo et al, 2012; Smallwood et al, 2011), and explore the influencing factors of visitor movement (Xia et al, 2011), broadening the microscale visitor mobility research content. TM also has economic, social, and cultural impacts on destinations through the movement of tourists. Numerous empirical studies have shown that tourist arrivals have a positive impact on economic growth (Pablo-Romero et al, 2013). Tourism is an important driver of economic growth (Tugcu, 2014). However, some studies have shown that tourist arrivals do not directly lead to economic growth, but promote TEG through regional economic development (Lee and Chien, 2008; Payne and Mervar, 2010; Odhiambo, 2011). The mobility of tourism will also bring about changes in destination transportation facilities. Transportation is not only an important carrier of TM, but also an important part of tourists' travel experience (Hannam et al, 2014). It also has a positive impact on destination company value together with TM (Zhang et al, 2020).

There are many theoretical discussions and empirical studies on the factors influencing TEG. From the perspective of suppliers, resource endowment (Melián-González and García Falcón , 2003; Wang, 2010; Zhu and He, 2019) and environmental quality (Katircioglu et al., 2014; Nitivattananon and Srinonil, 2019; Hamaguchi, 2021; Yong, 2021) are the fundamental factors determining tourism development. Simultaneously, as a typical service industry, human capital and physical capital in the tourism industry (Fahimi et al., 2018; Elsharnouby and Elbanna, 2021) and service level (Lin 2011) will impact tourism economic efficiency. From the perspective of demanders, the rise of per capita income and consumption upgrading continue to drive the transformation in the tourism industry (Feng and Sun, 2016), which in turn leads to an increasing scale of market demand (Fu et al., 2020), which provides the possibility of increasing the foreign exchange earnings, local capital accumulation, and consumption spillovers. From the perspective of supporters, scholars have verified the significant effects of factors on TEG, including the transportation facilities and accessibility (Macchiavelli and Pozzi, 2015; Khan et al., 2017; Zhong et al., 2019; Kanwal, 2020), the basis of the economy and marketization (Dritsakis, 2004), industrial structure (Zuo et al., 2020), public policy (Causevic and Lynch, 2013; Liu et al., 2020; Matteo, 2021), and technological progress (Sigala, 2018). 

In summary, the research on TM has paid attention to its impact on the regional economic, but they both ignored the role of TM on TEG.Studies of TEG based on static factors have primarily relied on econometric models (Tu and Zhang, 2020). Although the spatial spillover effects of influencing factors have gradually gained attention, its depth is limited and fails to explore the impact of TM and other related factors on the TEG. TM is becoming central to tourism activities and that understanding the capital mobility of tourism will have implications for tourism development under the new mobility paradigm(Sun et al. ,2016). This study proposes the concept of TM based on the theory of fluid mechanics, explores its impact on TEG, and analyzes the contribution of each influencing factor to TEG. 

(b) Econometric model

oA few words or a sentence should be sufficient to describe the spatial weight matrix used in equation 4.

Response b: Thank you very much for your reminder. The matrix we use in equation 4 is a adjacency matrix. We have supplemented it in section 2.2.3. 

(c) Data sources

oThe last paragraph of this section (about the seven regions of the study area) should be moved to the Appendix. Including a map inset of these areas would be helpful here, as well.

Response c: Thank you very much for your constructive comment. First, we moved the content of the seven regions of the study area to the Appendix as your suggestion. Second, we also added a map to more clearly present the geographical distribution of the study area.For your convenience, we present this section as below:

Supporting information

The study area is divided into seven regions according to geographical divisions of China. North China including Beijing, Tianjin, Hebei, Shanxi, Inner Mongolia. Northeast including Heilongjiang, Jilin, Liaoning. East China including Shanghai, Jiangsu, Zhejiang, Anhui, Jiangxi, Shandong, Fujian. Central China including Henan, Hubei, Hunan. South China including Guangdong, Guangxi, Hainan. Southwest China including Chongqing, Sichuan, Guizhou, Yunnan, Tibet. Northwest China including Shaanxi, Gansu, Qinghai, Ningxia, Xinjiang. Figure S shows the map of the seven regions.

Fig. S Map of the seven regions

(d) Spatial evolution

oIt would be helpful to include a LISA cluster map of to help readers better understand the spatial patterns described here.

Response d: Thank you very much for your constructive comment.We added the section 3.2.2 including LISA cluster map in page 14. For your convenience, we present this section as below:

3.2.2 Local Spatial Autocorrelation Cluster of Tourism Mobility

The global Moran's I cannot reflect the spatial correlation exhibited by local regions or individual provinces. We further use ArcGIS 10.8 to draw the LISA cluster diagram for 2000, 2005, 2010, 2015, and 2018 (Figure 4). The research samples are divided into four types of agglomeration: provinces with high TM are surrounded by provinces with high TM (H-H agglomeration), provinces with high TM are surrounded by provinces with low TM (H-L agglomeration), provinces with low TM are surrounded by provinces with high TM (L-H agglomeration), and provinces with low TM are surrounded by provinces with low TM (L-L agglomeration).

The results show that (1) provinces with H-H aggregation of TM in different periods are relatively stable; L-L and L-H aggregation types are stable but mixed with changes;The H-L aggregation type does not appear, which indicates that there is no "darkness under the light" area for China's provincial TM.Provinces with high TM can improve the TM of weekly provinces to a certain extent.(2) The H-H agglomeration is mainly concentrated in Jiangsu and Zhejiang. These regions are economically developed and have high per capita discretionary income. Moreover, the tourism infrastructure in these regions is more complete than taht in other regions, and the tourist reception scale is also higher, so their TM shows a high local concentration.(3) The L-L agglomeration types are mainly distributed in geographically remote areas such as Qinghai, Tibet, Gansu, and Xinjiang in inland China. Moreover, Xinjiang and Gansu temporarily withdraw from the L-L agglomeration area. The main reason for this pattern is that the transportation infrastructure in the areas above mentioned is relatively underdeveloped. The "space-time compression effect" brought about by the rapid development of China's transportation is not significant.Furthermore, due to the distance from the main tourist source markets, although the TM shows a high growth rate, it is still in the lowest-value area and the lower-value area for a long time.(4) L-H agglomeration is mainly transferred in Anhui, Shandong and Hebei, and these provinces are located in the “Leading Area”.The average value of TM in the surrounding provinces is generally high, forming a "collapse area" for TM.

Figure 4 LISA Clustering Results of TM

(e) Discussion

oThis section does not sufficiently analyze the authors’ findings in the context of previous studies. How and why does this study differ from previous research? Relatedly, there are some similarities between this study’s results and those of previous research—what is the unique contribution of this paper? Please emphasize these differences more clearly.

oIt seems that the policy implications of this manuscript are limited. Thus, the authors are encouraged to derive policy implications or recommendations based on their findings. Such implications need not be restricted to Chinese tourism-related policies but may also be applicable to other countries with conditions similar to China. The authors are encouraged to clarify these issues in this section.

oTheoretical and practical contributions should be highlighted more explicitly in this section. 

oThe authors are encouraged to expand their discussion of the research limitation(s), as this issue is currently insufficiently explained. 

Response e: Thank you very much for your constructive comment. We rewrite the Section “Discussion” and “Conclusion”.

First,we discuss and explain the findings of this study more fully and compare it with the existing literature in Section 5 Discussion. This section is presented as below:

5.1 Regression results of tourism mobility on tourism economic growth

This study briefly analyzes the regression results of the traditional and spatial panel data model. However, the spatial autocorrelation test results of TEG show an overall trend of fluctuating and increasing spatial correlation, especially with 2009 as the abrupt change point and a significant increase in the degree of agglomeration. Therefore, the article discusses the results of the spatial panel data model in detail, and the primary purpose of analyzing the traditional panel data model is to compare it with the spatial econometric results. 

The regression results of the spatial econometric model show that both TM and TA have a significant positive impact on TEG, which verifies the hypothesis we proposed above.This result is also consistent with Wu et al. (2012) and Perboli et al. (2015).In contrast, TP and TH have no significant impact on TEG.However, previous studies have also shown that the spatial spillover effect of tourism can significantly affect the TEG (Yang et al, 2012; Yang and Fik, 2014;Yuming, 2014).Therefore, the impact of TP and TH on TEG remains to be further confirmed. 

According to the decomposition results, TM will promote the growth of the local tourism economy but will have a negative impact on neighboring provinces, which indicates a more obvious competition in tourism development among provinces. The increase in mobility in a particular place under a given number of tourists will lead to a diversion of tourists, which will have a negative impact on neighboring regions.Therefore, the tourism industry should also pay attention to the competitive situation in the surrounding areas. The development of tourism focus not only on improving local tourism mobility but also on neighboring areas. Both TP and TH manifest substantial spatial spillover effects. The increase in TP and TH in neighboring areas will produce positive effects, making local areas attach importance to the development of tourism resources and enhancing tourism attraction. TA has a significant positive contribution to TEG, which is consistent with the conclusion of Yang et al, (2012).However, the spatial spillover effects of TA on TEG are not significant, which may be related to the fact that air traffic does not depend on adjacent spaces. 

5.2 Analysis of influencing factors’ contribution rate to tourism economic growth

TM and : The in North, Central, Southwest, and South China all show a trend of "falling and rising." It should be noted that the in North China was negative from 2005 to 2010, mainly due to the significant decline in TM in Tianjin and Hebei. The improvement in the transportation infrastructure has a significant impact on TM in Central and Southwest China. The opening of high-speed railroads is a fundamental reason for the fluctuation in . For South China, due to the implementation of the overnight visitor count statistics in the tourism statistics system of Guangdong in 2015~2018, the number of tourists decreased significantly compared to 2010~2015, which in turn led to a significant weakening of the . In contrast to the regions mentioned above, the in Northeast China shows a trend of "rising and falling" changes.From 2010 to 2015, the contribution of TM to TEG in Northeast China declined and was negative. The main reason is the overall decline of the regional economy in the Northeast region at this stage. In 2014 and 2015, the GDP growth rates of Northeast China were 4.23% and -0.84%, respectively, ranking second and last among the seven regions in China during the same period. .At the same time, the Northeast region began to carry out statistical "squeeze water" at this stage, which caused obvious fluctuations in the scale of tourists. Therefore, the downturn in the regional economic environment and stricter tourism statistics have negatively affected the contribution of tourism mobility to tourism economic growth. However, since 2016, China has put forward the " all-for-one tourism" policy. Provinces began to pay more attention to the role of tourism in regional economic growth. All-for-one tourism policies and new management systems have led to the continuous improvement of TM in Northeast China from 2015 to 2018, and the contribution to TEG has increased significantly compared with 2010-2015. The in East China gradually increased from 6.35% to 25.66%, which is related to the opening of the high-speed railroad network in 2010, leading to a significant increase in TM. Northwest China has made the tourism industry a key point for economic growth, and its tourist reception and transportation construction levels have been rapidly improved under the impetus of the all-for-one tourism strategy. 

Traffic and : The contribution of to TEG generally shows a downward trend. However, during the same period, Traffic showed a gradual upward trend. In 2018, it increased by 258.72% compared with 2000. Among them, it increased by 35.61% from 2000 to 2005, increased by 91.36% from 2005 to 2010, increased by 24.83% from 2010 to 2015, and increased by 10.73% from 2015 to 2018. From this, it can be judged that there may be a "threshold" in the transportation infrastructure. When the stock of transportation infrastructure in China reaches a certain level, the accumulation of transportation infrastructure cannot improve the contribution to the TEG.The role of transportation infrastructure in influencing tourists' decisions and determining tourist flow cannot be ignored. However, its contribution rate gradually decreases as transportation facilities are gradually improved and regional accessibility differences narrow. The is 14.82% during the examination period, in which the contribution rate of Traffic to TEG in East China (16.15%), Central China (17.44%), Southwest China (15.75%), and Northwest China (15.40%) is higher than that in North, Northeast and South China. This is mainly because Central China and East China are the regions with the largest passenger turnover in China. From 2000 to 2018, the average passenger turnover in Central China and East China was 118.988 billion person-kilometers and 84.595 billion person-kilometers, respectively. The Southwest China and Northwest China are among the regions with the fastest growth in passenger turnover in China, increasing by 3.13 times and 1.77 times respectively, ranking first and second in all regions.

Industry and : The tertiary industry consists of transportation, warehousing and postal industry, information transmission, real estate industry, financial industry, wholesale and retail industry, accommodation and catering industry, etc. Tourism is only a part of it. The per capita added value of the tertiary industry reflects the degree of development of the service industry in various regions, and this indicator has achieved a relatively large increase in terms of changing trends. It increased from 3,653 yuan in 2000 to 34,969 yuan, an increase of 8.57 times. The contribution of to TEG has gradually declined, mainly due to the slowdown in the growth rate of the per capita added value of the tertiary industry. The growth rate dropped from 91.30% in 2000-2005 to 34.35% in 2015-2018. The contribution of to TEG in North China, South China, Northwest China, and Southwest China is consistent with the national trend. Northeast China, East China, and Central China show different trends. Especially in the Northeast region, the contribution of to TEG has dropped significantly.The overall contribution rate of Industry reached 28.18%, indicating that the quality of tertiary industry development has a vital role in promoting TEG. is generally stable in East and Central China and declines significantly in Northeast China, which may be related to the deceleration of tertiary industry development, as the data show that the added-value of tertiary industry per capita in Liaoning, Heilongjiang, and Jilin increased by 93.04%, 75.15% and 90.43% from 2010 to 2015, while it only grew by 0.63%, 39.88% and 23.18% from 2015 to 2018.Central China was inconsistent with the overall national trend from 2005 to 2010. This is mainly due to the slow increase in the per capita added value of the tertiary industry during this period, ranking last in all regions. During this period, the industrial structure of Central China was still dominated by industry. In 2010, the average industrial added value accounted for 56.37% of GDP, the highest in all regions of the country. East China was inconsistent with the overall national trend in 2015-2018. The main reason is that the proportion of the tertiary industry in Fujian and Jiangxi in the region has not exceeded 50%, and there is a large room for optimization and improvement of the industrial structure. Therefore, the growth rate of the added value of the tertiary industry per capita exceeds the previous stage, and the contribution of to TEG is still rising. 

Structure and : The share of tertiary industry in tourism in Beijing and Tianjin increased significantly from 2010 to 2018 compared to 2000, leading to the rapid growth of in North China. The in Northeast China was -3.96% from 2005 to 2010, mainly since the growth rate of tertiary industry in Heilongjiang and Liaoning lagged behind that of the tourism industry. The in East, Central, and Southwest China is relatively stable, indicating that tourism and tertiary industry maintain a coordinated development. The in South China has achieved a shift from negative to positive growth. As the economic volume of Guangdong accounts for a large proportion in South China and the growth rate of tourism significantly lags behind the development rate of the tertiary industry, it leads to a low ∆S in South China from 2000 to 2010. The opening of high-speed rail provides new opportunities for tourism development, and the ∆S in South China gradually increased to 14.38% and 10.73% in 2010~2018. The ∆S in Northwest China has been increasing, which suggests that the tourism economy is the primary driver of tertiary industry growth. The continuous growth of the contribution to TEG is partially consistent with the findings of Chang et al. (2009), De Vita and Kyaw (2016), and Zuo and Huang (2017). The higher Structure is, the greater the contribution of to TEG. However, the literature above mentioned also pointed out that has a turning point. For example, Zuo and Huang (2017) found that this value in China is 8.25%.

Reception and : The has a negative impact on TEG.Zuo and Huang (2017) used the ratio of tourist arrivals to the permanent resident population to characterize tourism specialization in a study evaluating China's tourism-oriented economic growth. Before reaching the inflection point of 30.34 (that is, the tourism reception effect value is 0.03), this indicator has a significant positive impact on TEG. From 2000 to 2018, the tourism reception effect value dropped from 1.47 to 0.11, still less than 0.03. Therefore, the results of our study also partially confirm the research of Zuo and Huang (2017). While expanding the scale of tourists, various regions should also pay attention to the "inflection point" of the Reception value. When the inflection point is reached, the larger the scale of tourists is, the smaller the contribution to the TEG. However, the ratio of regional population to tourist decreases from 1.47 to 0.11 during the period from 2000 to 2018, indicating that not only the number of tourists should be taken into account, but also the quality of the tourism and the per capita tourism consumption should be attached importance to the TEG. The is relatively stable, among which the southwest and northwest China have the most significant negative contribution to the TEG, indicating that the growth rate of the number of tourists received in the above regions is higher than that of other regions. 

Second,we give more policy implications in section conclusion based on our findings. The revised part read as

Based on our findings above, we draw the following policy implications. To improve TEG, late-developing regions should improve TM by building large-scale tourism transportation infrastructure, promoting destination marketing to attract tourists, and paying attention to the possible negative effects of increased TM in neighboring regions.At the same time, the improvement of TM should be emphasized at different stages. The threshold effect of tourism transportation infrastructure should also be fully considered. After the transportation infrastructure reaches a certain stock, its contribution to TEG will decrease. At this time, expanding the scale of tourists should become the main tourism development policy.

Third, we highlight the theoretical and practical contributions more explicitly in section conclusion.Specifically, it read as

The main contributions of this study are as follows: (1) Based on fluid mechanics, we constructed an indicator of TM. We comprehensively consider the impact of tourist arrivals and transportation infrastructure on TEG, which is rarely proposed by scholars in the literature. Our research enriches the research on the influencing factors of TEG. (2) We analyze the influence of TM on TEG based on the econometric model, which highlights the importance of TM. Moreover, we found that TM has negative spatial overflow.(3) Based on the LMDI method, we decompose TEG into five major effects, rather than just considering traditional variables such as human input, capital input, and tourism resource input. Our study further enriches the research on the influencing factors of TEG.

Fourth, we expand the discussion of the research limitations. For your convenience, we present the paragraphs as below:

There are still some limitations in this study. It is difficult to directly collect data on the inflow and outflow of tourist between certain provinces. Therefore, we only select inflow of tourists as the primary data and do not consider the influence of the tourists’ outflow on TM. In fact, increased transport accessibility will not only expand the inflow of tourists but also affect the outflow of tourists. Therefore, the superposition effect of traffic and tourist inflow/outflow should be considered comprehensively to improve the scientific rationality of TM measurement. This study lacks comparative studies across multiple countries. The research in our study may show differentiated findings for developed or less developed countries.When constructing the econometric model, we mainly consider TM as the core explanatory variable, and only select human input and capital input, and air traffic related to traffic as control variables from the perspective of the economic growth model. In the future, the theory and practice of TM will be further explored with multivariate data to form a more rigorous and systematic cognitive framework.

We appreciate the detailed and constructive comments provided by the editor and all the reviewers, which have improved and refined this study. Thank you very much for consideration our manuscript.

Best regards,

The authors

References

Cárdenas-García, P. J., Sánchez-Rivero, M., & Pulido-Fernández, J. I. (2013). Does Tourism Growth Influence Economic Development? Journal of Travel Research, 54(2), 206–221. doi:10.1177/0047287513514297

Causevic S, Lynch P. (2013) Political (in) stability and its influence on tourism development. Tourism Management, 34: 145-157.

Chang, C.-L., T. Khamkaew, and M. J. McAleer. 2009. “A Panel Threshold Model of Tourism Specialization and Economic Development (No. EI 2009-40).” Erasmus School of Economics (ESE). http://hdl.handle.net/1765/17310 (accessed July 13, 2015).

Chen Y, Xie B, Zhang A. (2018) The impact of traffic on spatial mobility at different scales. Acta Geographica Sinica, 73(06):1162-1172. (in Chinese)

Chen, J., Yang, S., Li, H., Zhang, B., & Lv, J. (2013). Research on geographical environment unit division based on the method of natural breaks (Jenks). Int. Arch. Photogramm. Remote Sens. Spat. Inf. Sci, 3, 47-50.

Chew J. (1987) Transport and tourism in the year 2000. Tourism Management, 8(2): 83-85.

Chi, J. (2014). A cointegration analysis of bilateral air travel flows: The case of international travel to and from the United States. Journal of Air Transport Management, 39, 41-47.

Chong, C., Liu, P., Ma, L., Li, Z., Ni, W., Li, X., & Song, S. (2017). LMDI decomposition of energy consumption in Guangdong Province, China, based on an energy allocation diagram. Energy, 133, 525–544. doi:10.1016/j.energy.2017.05.045

Colinet Carmona, M. J., & Román Collado, R. (2015). LMDI decomposition analysis of energy consumption in Andalusia (Spain) during 2003–2012: the energy efficiency policy implications. Energy Efficiency, 9(3), 807–823. doi:10.1007/s12053-015-9402-y

Comerio N, Strozzi F. (2019) Tourism and its economic impact: A literature review using bibliometric tools. Tourism Economics. 25 (1): 109-131.

Connell, J., & Page, S. J. (2008). Exploring the spatial patterns of car-based tourist travel in Loch Lomond and Trossachs National Park, Scotland. Tourism Management, 29(3), 561–580. doi:10.1016/j.tourman.2007.03.019

De Vita, G., and K. S. Kyaw. 2016. “Tourism Specialization, Absorptive Capacity, and Economic Growth.” Journal of Travel Research. doi:10.1177/0047287516650042.

Dritsakis N. (2004) Tourism as a long-run economic growth factor: an empirical investigation for Greece using causality analysis. Tourism Economics. 10(3): 305-316.

Durden, G. C., & Silberman, J. (1975). The Determinant’s of Florida Tourist Flows: A Gravity Model Approach. Review of Regional studies, 5(3), 31-41.

Elsharnouby T H, Elbanna S. (2021) Change or perish: Examining the role of human capital and dynamic marketing capabilities in the hospitality sector. Tourism Management, 82:104184.

Fahimi A, Akadiri S S, Seraj M, et al. (2018) Testing the role of tourism and human capital development in economic growth. A panel causality study of micro states. Tourism Management Perspectives. 28: 62-70.

Fallah Ghalhari, G., & Dadashi Roudbari, A. (2018). An investigation on thermal patterns in Iran based on spatial autocorrelation. Theoretical and applied climatology, 131(3), 865-876.

Feng Q, Sun G. (2016) Effects of Per Capita GDP and Urbanization on Domestic Tourism Development in China’s Eight Regions. Areal Research and Development. 35(04):92-98.

Fonseca N, Sánchez-Rivero M. (2019) Significance bias in the tourism-led growth literature. Tourism Economics. 26(1): 137-154.

Fu X, Ridderstaat J, Jia H C. (2020) Are all tourism markets equal? Linkages between market-based tourism demand, quality of life, and economic development in Hong Kong. Tourism Management. 77: 104015.

Gunn, C. A. (1988). Vacationscape: Designing tourist regions. Van Nostrand Reinhold.

Haitovsky, Y., Salomon, I., & Silman, L. A. (1987). The economic impact of charter flights on tourism to Israel: An econometric approach. Journal of Transport Economics and Policy, 111-134.

Hallo, J. C., Beeco, J. A., Goetcheus, C., McGee, J., McGehee, N. G., & Norman, W. C. (2012). GPS as a Method for Assessing Spatial and Temporal Use Distributions of Nature-Based Tourists. Journal of Travel Research, 51(5), 591–606. doi:10.1177/0047287511431325

Hamaguchi Y. (2021) Does the trade of aviation emission permits lead to tourism-led growth and sustainable tourism? Transport Policy. 105: 181-192.

Hannam, K., Butler, G., & Paris, C. M. (2014). Developments and key issues in tourism mobilities. Annals of Tourism Research, 44, 171–185. doi:10.1016/j.annals.2013.09.010

Harman K, Sheller M, Urry J. (2006) Editorial: Mobilities, immobilities and moorings. Mobilities, 1 (1):1-22.

Hwang, Y. H., Gretzel, U., & Fesenmaier, D. R. (2006). Multicity trip patterns: Tourists to the United States. Annals of Tourism Research, 33(4), 1057-1078.

Kadir, N., & Karim, M. Z. A. (2012). Tourism and Economic Growth in Malaysia: Evidence from Tourist Arrivals from Asean-S Countries. Economic Research-Ekonomska Istraživanja, 25(4), 1089–1100.

Kanwal S, Rasheed M I, Pitafi A H, et al. (2020) Road and transport infrastructure development and community support for tourism: the role of perceived benefits, and community satisfaction. Tourism Management, 77: 104014.

Katircioglu S T, Feridun M, Kilinc C. (2014) Estimating tourism-induced energy consumption and CO2 emissions: the case of Cyprus. Renewable and Sustainable Energy Reviews. 29: 634-640.

Keum, K. (2010). Tourism flows and trade theory: a panel data analysis with the gravity model. The Annals of Regional Science, 44(3), 541-557.

Khadaroo J, Seetanah B. (2007) Transport infrastructure and tourism development. Annals of Tourism Research, 34 (4): 1021-1032.

Khadaroo, J., & Seetanah, B. (2008). The role of transport infrastructure in international tourism development: A gravity model approach. Tourism management, 29(5), 831-840.

Khan R, Abdul S, Dong Q, et al. (2017) Travel and tourism competitiveness index: The impact of air transportation, railways transportation, travel and transport services on international inbound and outbound tourism. Journal of Air Transport Management, 58:125-134.

Kim Y R, Williams A M, Park S, et al., (2021) Spatial spillovers of agglomeration economies and productivity in the tourism industry: The case of the UK. Tourism Management, 82: 104201.

Lee, C.-C., & Chien, M.-S. (2008). Structural breaks, tourism development, and economic growth: Evidence from Taiwan. Mathematics and Computers in Simulation, 77(4), 358–368. doi:10.1016/j.matcom.2007.03.004

Li LS, Yang F X, Cui C. (2019) High‐speed rail and tourism in China: An urban agglomeration perspective. International Journal of Tourism Research, 21 (1): 45-60.

Li, K. X., Jin, M., & Shi, W. (2018). Tourism as an important impetus to promoting economic growth: A critical review. Tourism Management Perspectives, 26, 135–142. doi:10.1016/j.tmp.2017.10.002

Li, X., Meng, F., & Uysal, M. (2008). Spatial pattern of tourist flows among the Asia-Pacific countries: An examination over a decade. Asia Pacific Journal of Tourism Research, 13(3), 229-243.

Lin L. (2011) The impact of service innovation on business performance: Evidence from firm-level data in Chinese tourism sector. IEEE: 1-5.

Liu A, Song H, Blake A. (2018) Modelling productivity shocks and economic growth using the Bayesian dynamic stochastic general equilibrium approach. International Journal of Contemporary Hospitality Management, 30 (11): 3229-3249.

Liu A, Wall G. (2006) Planning tourism employment: A developing country perspective. Tourism Management, 27 (1): 159-170.

Liu C, Dou X, Li J, et al. (2020) Analyzing government role in rural tourism development: An empirical investigation from China. Journal of Rural Studies, 79: 177-188.

Liu H, Song H, Wang Y. (2016) Inbound Tourism Demand and Economic Growth in China—Empirical Study Based on the Mixed Frequency Granger Causality Tests. 38(09):149-160. (in Chinese)

Liu Y, Shi J. (2017) How inter-city high-speed rail influences tourism arrivals: Evidence from social media check-in data. Current Issues in Tourism, 22(9): 1-18.

Liu, Y., & Han, Y. (2020). Factor structure, institutional environment and high-quality development of the tourism economy in China. Tourism Tribune, 35(3), 28-38.

Liu, Z., Lu, C., Mao, J., Sun, D., Li, H., & Lu, C. (2021). Spatial–Temporal Heterogeneity and the Related Influencing Factors of Tourism Efficiency in China. Sustainability, 13(11), 5825.

Lozano, S., & Gutiérrez, E. (2018). A complex network analysis of global tourism flows. International Journal of Tourism Research, 20(5), 588-604.

Lu, W., Liu, W., Hou, M., Deng, Y., Deng, Y., Zhou, B., & Zhao, K. (2021). Spatial–Temporal Evolution Characteristics and Influencing Factors of Agricultural Water use Efficiency in Northwest China—Based on a Super-DEA Model and a Spatial Panel Econometric Model. Water, 13(5), 632.

Lu, Y. (2022). The measurement of high-quality development level of tourism: Based on the perspective of industrial integration. Sustainability, 14(6), 3355.

Lue, C. C., Crompton, J. L., & Fesenmaier, D. R. (1993). Conceptualization of multi-destination pleasure trips. Annals of tourism research, 20(2), 289-301.

Macchiavelli A, Pozzi A. (2015) Low-cost flights and changes in tourism flows: evidence from bergamo-orio Al serio international. Tourism and Leisure, Springer Fachmedien, Wiesbaden: 323–336.

Massidda, C., & Etzo, I. (2012). The determinants of Italian domestic tourism: A panel data analysis. Tourism Management, 33(3), 603–610. doi:10.1016/j.tourman.2011.06.017

Masson S, Petiot R. (2009) Can the high speed rail reinforce tourism attractiveness? The case of the high speed rail between Perpignan (France) and Barcelona (Spain). Technovation, 29 (9): 611-617.

Matteo D D. (2021) Effectiveness of place-sensitive policies in tourism. Annals of Tourism Research, 19: 103146.

McKercher B. (1999) A Chaos approach to tourism. Tourism Management, 20 (4): 425-434.

Mckercher, B., & Lau, G. (2008). Movement patterns of tourists within a destination. Tourism geographies, 10(3), 355-374.

Melián-González A, García Falcón J M. (2003) Competitive potential of tourism in destinations. Annals of Tourism Research. 30 (3): 720-740.

Ministry of Culture and Tourism. (2020) Basic information of the tourism market in 2020. Beijing: Ministry of Culture and Tourism, PRC.

Morley, C., Rosselló, J., & Santana-Gallego, M. (2014). Gravity models for tourism demand: theory and use. Annals of tourism research, 48, 1-10.

Nazneen S, Xu H, Din N U. (2019) Cross‐border infrastructural development and residents' perceived tourism impacts: A case of China–Pakistan economic corridor. International Journal of Tourism Research, 21 (3): 334-343.

Neil L. (1990) Tourism Systems: An Interdisciplinary Perspective. Palmerston North, N.Z.: Department of Management Systems, Business Studies Faculty, Massey University.

Nitivattananon V, Srinonil S. (2019) Enhancing coastal areas governance for sustainable tourism in the context of urbanization and climate change in eastern Thailand. Advances in Climate Change Research. 10 (1): 47-58.

Nuno Carlos LeitÃ£o & Muhammad Shahbaz, 2016. "Economic Growth, Tourism Arrivals and Climate Change," Bulletin of Energy Economics (BEE), The Economics and Social Development Organization (TESDO), vol. 4(1), pages 35-43.

Odhiambo N.M. (2011). Tourism development and economic growth in Tanzania: Empirical evidence from the Ardl-bounds testing approach. Economic Computation & Economic Cybernetics Studies & Research, 45 (3), 71-83.

Oppermann, M. (1995). A model of travel itineraries. Journal of Travel Research, 33(4), 57-61.

Pablo-Romero, M. D. P., Sánchez-Braza, A., & Sánchez-Rivas, J. (2021). Tourism and electricity consumption in 9 European countries: a decomposition analysis approach. Current Issues in Tourism, 24(1), 82-97.

Pablo-Romero, M.d. P., & Molina, J. A. (2013). Tourism and economic growth: A review of empirical literature. Tourism Management Perspectives, 8, 28–41.

Payne, J. E., & Mervar, A. (2010). Research Note: The Tourism–Growth Nexus in Croatia. Tourism Economics, 16(4), 1089–1094. doi:10.5367/te.2010.0014

Pearce, D. G. (1979). Towards a geography of tourism. Annals of Tourism Research, 6(3), 245-272.

Perboli, G., Ghirardi, M., Gobbato, L., & Perfetti, F. (2015). Flights and their economic impact on the airport catchment area: an application to the Italian tourist market. Journal of Optimization Theory and Applications, 164(3), 1109-1133.

Prideaux B. (2000) The role of the transport system in destination development. Tourism Management, 21:53-63.

Prideaux, B. (2005). Factors affecting bilateral tourism flows. Annals of Tourism Research, 32(3), 780–801. doi:10.1016/j.annals.2004.04.008

Ren, H., Shang, Y., & Zhang, S. (2020). Measuring the spatiotemporal variations of vegetation net primary productivity in Inner Mongolia using spatial autocorrelation. Ecological Indicators, 112, 106108.

Saayman, A., & Botha, I. (2017). Non-linear models for tourism demand forecasting. Tourism Economics, 23(3), 594-613.

Saayman, A., & de Klerk, J. (2019). Forecasting tourist arrivals using multivariate singular spectrum analysis. Tourism Economics, 25(3), 330-354.

Shi, Z., Xu, D., & Xu, L. (2021). Spatiotemporal characteristics and impact mechanism of high-quality development of cultural tourism in the Yangtze River Delta urban agglomeration. PloS one, 16(6), e0252842.

Sigala M. (2018) New technologies in tourism: From multi-disciplinary to anti-disciplinary advances and trajectories. Tourism Management Perspectives, 25:151-155.

Sloane, A., & O’reilly, S. (2013). The emergence of supply network ecosystems: a social network analysis perspective. Production Planning & Control, 24(7), 621-639.

Smallwood, C. B., Beckley, L. E., & Moore, S. A. (2011). An analysis of visitor movement patterns using travel networks in a large marine park, north-western Australia. Tourism Management. doi:10.1016/j.tourman.2011.06.001

Song, H., & Li, G. (2008). Tourism demand modelling and forecasting—A review of recent research. Tourism management, 29(2), 203-220.

Song, H., & Witt, S. F. (2006). Forecasting international tourist flows to Macau. Tourism Management, 27(2), 214–224. doi:10.1016/j.tourman.2004.09.004

Sun J, Zhou S, Wang N, et al. (2016) Mobility in geographical research: Time, space and society. Geographical Research, 35(10):1801-1818.

Szivas, E., Riley, M., & Airey, D. (2003). Labor mobility into tourism. Annals of Tourism Research, 30(1), 64–76. doi:10.1016/s0160-7383(02)00036-1 

Tu J, Zhang D. (2020) Does tourism promote economic growth in Chinese ethnic minority areas? A nonlinear perspective. Journal of Destination Marketing & Management, 18: 100473.

Tugcu, C. T. (2014). Tourism and economic growth nexus revisited: A panel causality analysis for the case of the Mediterranean Region. Tourism management, 42, 207-212.

Urry J. (2003): Global Complexity. Cambridge: Polity, 101.

Wang Y. (2010) Tourism resource endowment and regional tourism economies: A positive analysis on Shanxi Province. Ecological Economy. 8: 41-45.

Wang, D., Wang, L., Chen, T., Lu, L., Niu, Y., & Alan, A. L. (2016). HSR mechanisms and effects on the spatial structure of regional tourism in China. Journal of geographical sciences, 26(12), 1725-1753.

Wang, X., Huang, S., Zou, T., & Yan, H. (2012). Effects of the high speed rail network on China’s regional tourism development. Tourism Management Perspectives, 1, 34–38. doi:10.1016/j.tmp.2011.10.001

Wu, C., Hayashi, Y., & Funck, C. (2012). The role of charter flights in Sino-Japanese tourism. Journal of Air Transport Management, 22, 21-27.

Xia, J. (Cecilia), Zeephongsekul, P., & Packer, D. (2011). Spatial and temporal modelling of tourist movements using Semi-Markov processes. Tourism Management, 32(4), 844–851. doi:10.1016/j.tourman.2010.07.009

Yang, Y., & Fik, T. (2014). Spatial effects in regional tourism growth. Annals of Tourism Research, 46, 144–162. doi:10.1016/j.annals.2014.03.007

Yang, Y., & Wong, K. K. F. (2012). A Spatial Econometric Approach to Model Spillover Effects in Tourism Flows. Journal of Travel Research, 51(6), 768–778. doi:10.1177/0047287512437855

Yong E L. (2021) Understanding the economic impacts of sea-level rise on tourism prosperity: Conceptualization and panel data evidence. Advances in Climate Change Research. https://doi.org/10.1016/j.accre.2021.03.009

Yuming, W. (2014). Spatial Panel Econometric Analysis of Tourism Economic Growth and its Spillover Effects. Tourism Tribune/Lvyou Xuekan, 29(2).

Zha, J., Dai, J., Ma, S., Chen, Y., & Wang, X. (2021). How to decouple tourism growth from carbon emissions? A case study of Chengdu, China. Tourism Management Perspectives, 39, 100849.

Zhang, A., Liu, L., & Liu, G. (2020). High-speed rail, tourist mobility, and firm value. Economic Modelling. doi:10.1016/j.econmod.2020.05.004

Zhang, Y., Li, X., & Wu, T. (2019). The impacts of cultural values on bilateral international tourist flows: a panel data gravity model. Current Issues in Tourism, 22(8), 967-981. 

Zheng, W., Huang, X., & Li, Y. (2017). Understanding the tourist mobility using GPS: Where is the next place? Tourism Management, 59, 267–280. doi:10.1016/j.tourman.2016.08.009

Zhong L. Sun S. Law R. (2019) Movement patterns of tourists. Tourism Management. 75:318-322.

Zhu X Y, He Y H. (2019) Does tourism promote economic growth in the ethnic areas of China? Emerging Markets Finance and Trade. 57(2): 386-399.

Zuo B, Cai S, Yang Y, et al. (2020) The structural changes of topological network and its effects on performance in tourism industry: A case of Yangshuo, China. Tourism Tribune, 35(6): 25-39.

Zuo, B., & Huang, S. (Sam). (2017). Revisiting the Tourism-Led Economic Growth Hypothesis: The Case of China. Journal of Travel Research, 57(2), 151–163. doi:10.1177/0047287516686725

---

## [Decision Letter · Decision Letter 1]

20 Sep 2022

The Contribution of Tourism Mobility to Tourism Economic Growth in China

PONE-D-22-06360R1

Dear Dr. Yu,

We’re pleased to inform you that your manuscript has been judged scientifically suitable for publication and will be formally accepted for publication once it meets all outstanding technical requirements.

Kind regards,

Hironori Kato, Dr. Eng.

Academic Editor

PLOS ONE

**Comments to the Author**

1. If the authors have adequately addressed your comments raised in a previous round of review and you feel that this manuscript is now acceptable for publication, you may indicate that here to bypass the “Comments to the Author” section, enter your conflict of interest statement in the “Confidential to Editor” section, and submit your "Accept" recommendation.

Reviewer #1: All comments have been addressed

Reviewer #2: All comments have been addressed

2. Is the manuscript technically sound, and do the data support the conclusions?

Reviewer #1: (No Response)

Reviewer #2: Yes

3. Has the statistical analysis been performed appropriately and rigorously? 

Reviewer #1: (No Response)

Reviewer #2: Yes

4. Have the authors made all data underlying the findings in their manuscript fully available?

Reviewer #1: (No Response)

Reviewer #2: No

5. Is the manuscript presented in an intelligible fashion and written in standard English?

Reviewer #1: (No Response)

Reviewer #2: Yes

6. Review Comments to the Author

Reviewer #1: In response to my comments in the previous review, the authors have rewritten the paper. For the part I reviewed, it seems improved. I am happy with this revised version.

Reviewer #2: (No Response)

7. PLOS authors have the option to publish the peer review history of their article (what does this mean?). If published, this will include your full peer review and any attached files.

Reviewer #1: No

Reviewer #2: No

---

## [Editor Report · Acceptance letter]

19 Oct 2022

PONE-D-22-06360R1 

The contribution of tourism mobility to tourism economic growth in China 

Dear Dr. Yu:

I'm pleased to inform you that your manuscript has been deemed suitable for publication in PLOS ONE. Congratulations! Your manuscript is now with our production department. 

Kind regards, 

on behalf of

Dr. Hironori Kato 

Academic Editor

PLOS ONE